# Long-Term Fertilization and Lime-Induced Soil pH Changes Affect Nitrogen Use Efficiency and Grain Yields in Acidic Soil under Wheat-Maize Rotation

Nano Alemu Daba [1,†], Dongchu Li [1,2,†], Jing Huang [1,2], Tianfu Han [1], Lu Zhang [1,2], Sehrish Ali [1], Muhammad Numan Khan [1], Jiangxue Du [1], Shujun Liu [1,2], Tsegaye Gemechu Legesse [1], Lisheng Liu [1,2], Yongmei Xu [3], Huimin Zhang [1,2,*] and Boren Wang [1,2,*]

1   National Engineering Laboratory for Improving Quality of Arable Land, Institute of Agricultural Resources and Regional Planning, Chinese Academy of Agricultural Sciences, Beijing 100081, China; nanoalemu2001@gmail.com (N.A.D.); lidongchu@caas.cn (D.L.); huangjing@caas.cn (J.H.); hantianfu123@126.com (T.H.); zhanglu01@caas.cn (L.Z.); sehrishbakhsh788@outlook.com (S.A.); khanmn123@outlook.com (M.N.K.); dujiangxue@126.com (J.D.); liushujun@caas.cn (S.L.); tsegayeccd@gmail.com (T.G.L.); liulisheng@caas.cn (L.L.)
2   National Observation Station of Qiyang Agri-Ecology System, Institute of Agricultural Resources and Regional Planning, Chinese Academy of Agricultural Sciences, Qiyang 426182, China
3   Institutes of Soil, Fertilizer and Agricultural Water Conservation, Xinjiang Academy of Agricultural Sciences, Urumqi 830091, China; xym1973@163.com
*   Correspondence: zhanghuimin@caas.cn (H.Z.); wangboren@caas.cn (B.W.)
†   Nano Alemu Daba and Li Dongchu equally contributed to this article.

**Abstract:** Liming (L) is a common practice to mitigate soil acidification and enhance soil quality and crop productivity. However, in acidic soil, it is not clear how long-term application of lime and nitrogen (N)-based fertilizer affects soil chemical properties, the wheat and maize grain yields (GY), and N-use efficiency (NUE). Thus, to investigate the effects of N-based fertilizations without L (−L) and with L (+L) on wheat and maize GY and NUE through their effects on soil chemical properties, we analyzed a 28-year field experiment in acidic soil under a wheat-maize system in South China. The analysis was carried out between 1991 and 2010 (before L) and between 2011 and 2018 (after L). We categorized the treatments into (1) no fertilizer (C); nitrogen (N); N and phosphorus (NP); N and potassium (NK); N, P and K (NPK); and NPKC$_R$, NPK and crops residue (C$_R$) applications (NPKC$_R$), before L; and (2) C; N−L; N+L; NP−L; NP+L; NK−L; NK+L; NPK−L; NPK+L; NPKC$_R$−L and NPKC$_R$+L, after L. The effects of long-term fertilization resulted in lower soil pH by 15%, soil available K (AK) by 19%, P$_{Olsen}$ by 6%, NO$_3^-$-N by 15%, soil organic matter (SOM) by 16%, total N by 16%, and C:N ratio by 13% in −L soil than in +L soil. However, the accumulation of NH$_4^+$-N was higher by 40% in −L soil than in +L soil. Wheat and maize GY, N recovery efficiency (RE$_N$), and N partial factor productivity (PEP$_N$) were more adversely affected by 8-year fertilizations in −L compared with fertilizations before L and in +L primarily because of the significantly decreased soil pH. Conversely, improvements in wheat and maize yields, RE$_N$, and PFP$_N$ by 8-year fertilizations in +L were related to increasing soil pH, exchangeable base cations such as Ca$^{2+}$, Mg$^{2+}$, and the alleviated toxicity of Al$^{3+}$. Overall, improvement of GY and NUE from the acidified soil in South China requires the long-term integrated use of fertilizer (NPK), retention of C$_R$, and the +L (i.e., NPKC$_R$+L).

**Keywords:** fertilization with liming; fertilization without liming; N partial factor productivity; N recovery efficiency; soil acidity

## 1. Introduction

It is widely recognized that in different parts of the world, soil and crop productivity are losing profitability and sustainability, mainly due to intensive agriculture and the lack of

improved integrated management practices [1–3]. In China, fertilizer application has been the most common agricultural practice used to improve soil fertility and crop productivity in the last three decades [2,3]. However, continuous and excessive fertilizer application, particularly urea as a source of nitrogen (N), poses environmental challenges such as increased soil acidification, eutrophication, and nitrate pollution of bodies of water [4,5].

Soil acidification, which indicates the relative acidic and basic cations distributions [6,7], has become a major crop production constraint in the dryland production system of red soil in South China [8,9]. Acidic soil with pH < 5.5, directly affects crop growth through acidic reactions and shows indirect effects on crop growth by affecting nutrient availability [10]. Common acid-forming cations include hydrogen ($H^+$), aluminum ($Al^{3+}$) and iron ($Fe^{2+}$ or $Fe^{3+}$), whereas calcium ($Ca^{2+}$), magnesium ($Mg^{2+}$), potassium ($K^+$), and sodium ($Na^+$) are the common base-forming cations. Changes in soil pH induced by N fertilization in acidic soil affects the soil biochemical properties, and the availability of nutrients in the soil; hence, the growth of the plant and the general ecosystem functionality as well [11–13]. In other words, when soil pH is below 5.5 in acidic soil, N, phosphorus (P), K, sulfur (S), Ca, Mg, and molybdenum (Mo) are deficient in acid soils and a pH increase is required to obtain optimal growing conditions for crops [14].

However, the dynamic changes in the soil's pH over time greatly depend on the source and type of the soil acidity amendment materials and the fertilizers used and their interaction [15,16]. Amendment of soil acidity through liming (L) is one of the most effective strategies used to mitigate soil acidification [17,18]. This is because L plays a vital role in regulating soil pH and the toxicity instigated due to H, Al, and Mn [19,20]. It also improves the soil's physico-chemical properties and the plants' nutrient uptake [21,22]. This is because L produces bicarbonate ($HCO_3^-$) and hydroxide ($OH^-$), the latter takes $H^+$ and $Al^{3+}$ (acid-forming cations) out of the solution, which in turn, increases the soil pH by reacting with carbon dioxide and water. Furthermore, the duration of L has a great impact on the effectiveness of L in improving the productivity of acidic soil. Long-term L, for instance, contributes significantly to soil pH change, nutrient concentration, and grain yield (GY) in acidic soil compared with short-term L [23,24]. Yet, what remains to be observed is the mechanism by which long-term fertilization and L-induced soil pH changes affect crop N uptake, NUE, and GY. Nevertheless, it is generally agreed that crop residue ($C_R$) incorporation is important in improving soil nutrient distribution and crop productivity [25] because it increases the extent of soil nutrients available to the crops [26–28] through initially immobilizing and then gradually mineralizing N [29–32]. However, there is a lack of adequate study that elucidates the effects of a $C_R$ addition along with other fertilizer applications, such as N, P, and K plus lime on the soil chemical properties, crop N uptake, and NUE in South China.

Therefore, in this study, we define the NUE of wheat-maize systems based on its two commonly known indices [33–35]. The first is N recovery efficiency ($RE_N$), which is the proportion of N fertilizer that is removed through the harvested crop biomass during the growing season. The second is N partial factor productivity ($PFP_N$), which means the ratio of crop yield is per unit of applied N fertilizer. Previous reports from long-term studies have shown relatively low values of $RE_N$ (16–18%) and $PFP_N$ (31–38 kg kg$^{-1}$) in the Chinese intensive wheat-maize system [36]. Therefore, it is believed that NUE improvements in crop production will have important economic and environmental benefits to China's agriculture [37]. Likewise, in China, adequate fertilization and L of acidic soils are believed to be essential for sustainable maize and wheat productivity when used with high $RE_N$ and $PFP_N$, mainly because doing this ensures good soil properties, especially amelioration of soil pH [36].

However, long-term field experiments have not been conducted on the comprehensive impacts of the co-application of L and N-based fertilizers on soil pH dynamics, yield, and NUE in South China, where winter wheat and summer maize are cultivated as important rotational crops. Previous studies in this area mainly focused on the long-term fertilization effects on soil quality and crops productivity [38–41]. Most recently, however, few studies

determined how long-term fertilization and L affected the links between soil exchangeable cations and K-uptake [42] and P-use efficiency [43] in this area. Therefore, further studies are needed to understand how soil properties, yield, and NUE respond to the co-application of L and N-based fertilizers under the wheat-maize rotation in South China. We hypothesized that the L-induced soil pH changes positively and strongly correlate with crop GY and $RE_N$, and $PFP_N$. This hypothesis was tested by evaluating the impact of long-term N-based fertilizations without liming (−L) and with liming (+L) on (i) wheat and maize yields, (ii) $RE_N$, and $PFP_N$ (iii) soil pH. In this study, our intent is to investigate the effects of 28-year fertilization and an 8-year L on GY, and NUE through their effects on soil pH changes under a wheat-maize rotation system in acidic soil in South China.

## 2. Materials and Methods

### 2.1. Description of the Study Site

A long-term field fertilization experiment was established in 1990 for a wheat-maize rotation system at the Chinese National Observation and Research Station, Qiyang, Hunan Province. The site is located at 120 m above sea level, 26°45′ N latitude and 111°52′ E longitude. The area has a warm subtropical humid monsoon climate, with a mean annual rainfall and temperature of 1290 mm and 17.8 °C, respectively. The soil at the experimental site is a Eutric Cambisol and red soil with high clay content (43.9%) [44]. The detailed climatic and soil properties information of this site were provided by [5,43].

### 2.2. Experimental Design and Procedures

For this study, we addressed 28 total years of a field experiment in two split long-term periods in the wheat-maize rotation system (Figure S1).

Firstly, from 1991 to 2010 (before L), six fertilizer treatments were arranged in a randomized block design with two replicates with an area of 200 m$^2$ (20 m by 10 m) for each plot. Adjacent plots were separated from each other by 20 cm cemented baffle plates to avoid the water and treatment contamination from the nearby plot. The fertilizer treatments consisted of: (1) no fertilizer application or control (C); (2) nitrogen (N); (3) N and phosphorus (NP); (4) N and potassium (NK); (5) N, P, and K (NPK); (6) NPK plus crop residue (NPKC$_R$). Based on the initial soil test in 1990 for soil nutrient status and crop nutrient requirement, the entire fertilizers of N, P, and K were applied at the rate of 150 kg N ha$^{-1}$ year$^{-1}$, 120 kg P$_2$O$_5$ ha$^{-1}$ year$^{-1}$, and 120 kg K$_2$O ha$^{-1}$ year$^{-1}$ within the rows at sowing. Annually, 30% and 70% of all fertilizer inputs were assigned to the wheat and maize crop, respectively. The sources of N, P, and K were urea, calcium super-phosphate, and potassium chloride, respectively. We referred to the super-phosphate application as a P treatment because sulfur (S) deficiency in China is low due to the use of S-containing fertilizers and the occurrence of S input from the rainfall. Because of the high C:N ratio of crop straw, only half of the aboveground biomass of the crops were incorporated into the 0–20 cm soil layer before the wheat was planted, and one month after, maize was sown under the NPKC$_R$ treatment. Before it was incorporated into the soil, the crop straw was manually chopped into small pieces of 3 cm lengths. The additional nutrients input through the C$_R$ incorporation were not considered in this study.

Secondly, due to severe soil acidification, all treatments mentioned in before L were split into two equal demi-plots, each with an area of 20 m by 4.9 m in 2010 (Figure S1). Then, from 2011 to 2018 (after L), a demi-plot of each treatment was amended with quicklime (CaO), which resulted in without L (−L) and with L (+L) sub-plots. Thus, the eleven (11) treatments: (1) C; (2) N−L; (3) N+L; (4) NP−L; (5) NP+L; (6) NK−L; (7) NK+L; (8) NPK−L; (9) NPK+L; (10) NPKC$_R$−L; (11) NPKC$_R$+L, were arranged in a split-plot design with fertilizer treatments to the main plots and L to the sub-plots. The L rate was 2550 kg ha$^{-1}$ in 2011 and 1500 kg ha$^{-1}$ in 2014, to remediate the soil acidification where the fertilizer treatments were managed as before L. The dosage of the lime application was calculated based on the initial soil pH in 2010 and soil buffer capacity and incorporated into the soil depth of 15 cm. Here, we addressed two issues: (i) how different N-based fertilizations,

−L and +L, influence the soil chemical properties, wheat and maize GY, and NUE, during the period of 2011–2018, and (ii) how the dynamic changes in soil pH under −L and +L related with other soil chemical properties, such as $RE_N$, $PFP_N$, and GY.

Each year, sampling units within a treatment plot were grouped into 2 strata before a set of subsampling units was selected randomly from each stratum using a stratified random sampling technique [45]. Hence, the total number of sampling units per treatment was 4 (2 reps × 2 subsampling units), and the total number of sampling units per experiment before L was 24 (2 reps × 2 subsampling units × 6 treatment plots) and after L was 44 (2 reps × 2 subsampling units × 11 treatment plots).

The experimental field was cultivated by a rot tiller to the depth of 25–30 cm. Then a field layout was prepared following the winter wheat (*Triticum aestivum* L.)-summer maize (*Zea mays* L.) rotation specification design. During the experiment, four varieties of wheat (Xiaoyan 6, Laizhou 953, Shan 253, and Xiaoyan 22) and four maize varieties (Shandan 9, Hudan 4, Gaonong 1, and Zhengdan 958) were changed every 5 years. Sowing was carried out in October for the winter wheat and summer maize, the wheat was harvested in June and the maize was harvested from late September to early October.

### 2.3. Soil Sampling and Analysis

In each year, five soil samples from 0 to 20 cm and 20 to 40 cm were collected within two weeks of winter wheat post-harvest, using an auger for each treatment. The five samples were bulked together, and thoroughly mixed. Then the composite samples were air-dried and ground using a pestle and mortar and passed through a 1 mm sieve. The pH of soil was determined with a glass electrode at a water-soil suspension ratio of 2.5:1, according to a pHS−3C mv/pH detector in Shanghai, China. $P_{Olsen}$ was extracted using 0.5 M sodium bicarbonate tumbled for 30 min [46]. Phosphorus concentration in the extracted solution was measured using the Mo-Sb spectrophotometry method [46]. Soil available K (AK) was extracted using an $NH_4COOCH_3$ solution and determined with flame photometry (FP640, Shanghai, China). Organic matter content was determined by the $K_2Cr_2O_7$ oxidation method [47]. The soil nitrate-N ($NO_3^-$-N) and ammonium-N ($NH_4^+$-N) were determined after 1 mol $L^{-1}$ KCl (at a ratio of 1:10 (*w/v*)) soil solution extraction by a continuous flow analyzer (San$^{++}$, Skalar, Holland). The total N (TN) was analyzed by the Kjeldahl method [48]. Exchangeable $Ca^{2+}$ and $Mg^{2+}$ were extracted with 1 mol ammonium acetate $L^{-1}$ (pH 7), and exchangeable $Al^{3+}$ was determined with 0.1 mol $BaCl_2$ $L^{-1}$ and then was analyzed by an atomic absorption spectroscopy procedure.

### 2.4. Plant Sampling

Plant samples were collected from the center of each plot at crop maturity, and they were separated into stems, leaves and chaffs to determine the N contents in the grain and straw. Samples were then oven-dried at a temperature of 80 °C for over 24 h to a constant dry weight. The samples were ground using a rotor mill and allowed to pass through 2 mm and 0.5 mm sieves for the straw and the grain, respectively. The N content (in%) in the plant tissue was determined using the micro-Kjeldahl digestion method as described in the guide for a plant nutrient analysis [49]. The carbon content was determined from pulverized samples by the dry-combustion method using an elemental analyzer (Elementar Analysensysteme GmbH, Hanau, Germany). The GY and straw in kg ha$^{-1}$ were determined after the air-dried wheat and maize crops were harvested, and their yields were adjusted to 12% moisture content.

### 2.5. Calculations

Grain and straw N uptake (kg ha$^{-1}$) of the crops was calculated as grain and straw yield multiplied in each plot by their respective percentage of N content of grain and straw. The total N (kg ha$^{-1}$) uptake was computed as the sum of grain N uptake and straw N

uptake. The total N uptake in the straw and grain samples were used to evaluate the $RE_N$ and $PFP_N$ of wheat and maize [50,51], and then calculated according to [33] as follows:

$$RE_N \left(kg\ kg^{-1}\right) = (UN - UN_0)/FN \times 100 \tag{1}$$

$$PFP_N \left(kg\ kg^{-1}\right) = GY/FN \tag{2}$$

where UN is the total N uptake ($kg\ ha^{-1}$) in the fertilized plot, $UN_0$ is the total N uptake ($kg\ ha^{-1}$) in the control plot, and FN is the amount of N applied to the plot. The designation GY is the grain yield ($kg\ ha^{-1}$) in the fertilized plot.

### 2.6. Statistical Analysis

The effects of fertilizer, liming (L), and fertilizer $\times$ L and the year on soil properties, GY and NUE were tested with a linear mixed model (LMM) at the 0.05 level in Genstat$^\circledR$ [52]. Treatments, i.e., fertilizer (before L) and fertilizer $\times$ L (after L) were treated as fixed effects, whereas replicates within samples in each treatment plot and year (included as a repeated measure) were fitted as random effects. When the LMM showed the presence of significance at $p \leq 0.05$ for treatments, analysis of variance by REML was used to analyze the fixed effects of treatments on response variables, using the blocking structures of rep/fertilizer/subsample/year and rep/fertilizer/lime/subsample/year before and after L, respectively. The present difference between treatments was determined at least significant difference (LSD) $p \leq 0.05$ in the tables and figures.

The linear mixed model used for the analysis of data before L was applied is given as follows:

$$Y_{ijkl} = \mu + \alpha_i + \beta_j + \rho_{(ij)k} + \Upsilon_{(k)l} + \epsilon_{ijkl} \tag{3}$$

where $Y_{ijkl}$ is the response variables (soil chemical properties, GY, N uptake and NUE) obtained over the period of 1991–2010 from fertilizer treatments (C, N, NP, NK, NPK and $NPKR_C$); $\mu$ is the overall mean; $\alpha_i$ is the random effect of replication; $\beta_j$ is the fixed effect of fertilization; $\rho_{(ij)k}$ is the random sample effect within replications per a treatment plot; $\Upsilon_{(k)l}$ is the random effect of 20 years; and $\epsilon_{ijkl}$ is the random error.

The linear mixed model used for the analysis of data after L was applied is given as follows:

$$Y_{ijklm} = \mu + \alpha_i + \delta_j + \beta_k + \delta\beta_{jk} + \rho_{(jk)l} + \Upsilon_{(jk)m} + \epsilon_{ijklm} \tag{4}$$

where $Y_{ijklm}$ is the response variables (soil chemical properties, GY, N uptake and NUE) obtained over the period of 2011–2018 from fertilizer treatments (C, N, NP, NK, NPK and NPKRC) at two levels of liming (−L and +L) and their interaction; $\mu$ is the overall mean; $\alpha_i$ is the random effect of replication; $\delta_j$ is the fixed effect of liming; $\beta_k$ is the fixed effect of fertilization; $\delta\beta_{jk}$ is the interaction fixed effect of fertilization and liming; $\rho_{(jk)l}$ is the random sample effect within replications per a treatment plot; $\Upsilon_{(jk)m}$ is the random effect of 8 years; and $\epsilon_{ijklm}$ is the random error.

Moreover, the relationship between soil pH, exchangeable soil cations ($Al^{3+}$, $Ca^{2+}$ and $Mg^{2+}$), and years after L were quantified by the linear regression equation. A redundancy analysis (RDA) was performed to evaluate the correlation between soil chemical properties ($NO_3^--N$, $NH_4^+-N$, pH, SOM $Ca^{2+}$ and $Al^{3+}$) and GY, $RE_N$ and $PEF_N$ using CANOCO for Windows (version 5) in the long-term liming and fertilization.

## 3. Results

### 3.1. Soil Chemical Properties

The results of soil chemical properties in the surface soil (0–20 cm), before and after L experiments are presented here under Tables 1 and 2, respectively. The results show that the soil pH, $NO_3^--N$, SOM, TN and C:N ratio were more strongly affected by 8 years of liming than 20 years of fertilizations.

**Table 1.** Soil chemical properties of the 0–20 cm soil layer as a function of the applied fertilization. Average values of 1991–2010 (before liming).

| Fertilizer | pH | AK (mg kg$^{-1}$) | P$_{Olsen}$ (mg kg$^{-1}$) | NO$_3^-$-N (mg kg$^{-1}$) | NH$_4^+$-N (mg kg$^{-1}$) | SOM (g kg$^{-1}$) | TN (g kg$^{-1}$) | C:N Ratio |
|---|---|---|---|---|---|---|---|---|
| C | 5.7 a | 60.0 b | 0.7 b | 4.4 a | 0.5 c | 14.7 a | 1.1 a | 12.8 a |
| N | 4.7 a | 59.9 b | 0.6 b | 3.6 a | 3.1 a | 19.0 a | 1.3 a | 9.4 a |
| NP | 4.9 a | 64.0 b | 109.4 a | 4.6 a | 1.6 b | 22.0 a | 1.4 a | 10.6 a |
| NK | 4.7 a | 256.9 a | 0.7 b | 4.3 a | 1.7 b | 20.9 a | 1.4 a | 11.2 a |
| NPK | 5.0 a | 283.2 a | 116.4 a | 4.5 a | 1.6 b | 20.8 a | 1.5 a | 11.7 a |
| NPKC$_R$ | 5.1 a | 276.7 a | 121.8 a | 4.5 a | 1.6 b | 24.7 a | 1.5 a | 11.8 a |
| LSD ($p \leq 0.05$) | ns | 46.83 | 20.34 | ns | 0.51 | ns | ns | ns |

Means followed by different letters within the same column are significantly different from each other at $p \leq 0.05$ by LSD test with ns denoting $p > 0.05$. C: no fertilizer application or control, N: nitrogen, NP: N and phosphorus (P), NK: N and potassium (K), NPK: N, P, and K, NPKC$_R$: NPK plus crop residue (CR), AK: soil available K, SOM: soil organic matter, and TN: soil total N.

**Table 2.** Soil chemical properties of the 0–20 cm soil layer as a function of the applied fertilization. Average values of 2011–2018 (after liming).

| Treatment Lime | Fertilizer | pH | AK (mg kg$^{-1}$) | P$_{Olsen}$ (mg kg$^{-1}$) | NO$_3^-$-N (mg kg$^{-1}$) | NH$_4^+$-N (mg kg$^{-1}$) | SOM (g kg$^{-1}$) | TN (g kg$^{-1}$) | C:N Ratio |
|---|---|---|---|---|---|---|---|---|---|
| Without lime(−L) | C | 5.8 a | 45.3 d | 0.6 d | 4.1 de | 0.5 g | 16.1 e | 0.99 c | 11.8 b |
| | N | 4.3 e | 45.6 d | 0.5 d | 3.3 f | 3.8 a | 11.4 f | 0.99 c | 8.6 g |
| | NP | 4.5 de | 53.6 d | 62.4 c | 4.2 de | 1.8 cd | 17.4 de | 1.05 c | 9.1 fg |
| | NK | 4.3 e | 220.9 c | 0.6 d | 3.9 e | 2.5 b | 16.3 e | 0.98 c | 9.9 ef |
| | NPK | 4.5 de | 239.3 b | 77.5 a | 4.3 cde | 1.7 de | 20.1 bc | 1.04 c | 10.3 de |
| | NPKC$_R$ | 4.6 de | 236.6 b | 79.1 a | 4.3 cde | 1.6 def | 18.2 d | 1.06 bc | 11.3 bc |
| With lime(+L) | N | 4.8 cd | 56.1 d | 0.7 d | 4.4 bcd | 1.9 c | 16.4 e | 1.18 a | 10.0 def |
| | NP | 5.2 bc | 57.1 d | 70.1 b | 4.7 ab | 1.6 ef | 18.9 cd | 1.16 ab | 10.6 cde |
| | NK | 4.8 cd | 239.4 b | 0.7 d | 4.4 bcd | 1.6 def | 18.6 cd | 1.18 a | 10.4 cde |
| | NPK | 5.3 b | 262.1 a | 80.5 a | 4.6 abc | 1.6 ef | 21.9 a | 1.19 a | 11.0 bcd |
| | NPKC$_R$ | 5.3 b | 274.4 a | 82.2 a | 4.8 a | 1.5 f | 20.8 ab | 1.24 a | 13.9 a |
| LSD ($p \leq 0.05$) | | 0.44 | 15.26 | 4.79 | 0.39 | 0.18 | 1.61 | 0.10 | 0.99 |

Means followed by different letters within the same column are significantly different from each other at $p \leq 0.05$ by LSD test with ns denoting $p > 0.05$. C: no fertilizer application or control, N: nitrogen, NP: N and phosphorus (P), NK: N and potassium (K), NPK: N, P, and K, NPKC$_R$: NPK plus crop residue (CR), AK: soil available K, SOM: soil organic matter, and TN: soil total N.

Before L (Table 1), AK, P$_{Olsen}$ and NH$_4^+$-N were significantly ($p \leq 0.05$) influenced by the 20 years of fertilizations, with the highest accumulation of AK for the NPK, NPKC$_R$ and NK treatments, and P$_{Olsen}$ for the NPKC$_R$, NPK and NP, while highest NH$_4^+$-N occurred for the N-alone treatment. These revealed that the application of K fertilizers increased AK while the application of P fertilizer increased P$_{Olsen}$.

After L (Table 2), the contents of selected soil chemical properties were generally higher under fertilizations +L than its equivalent under fertilizations −L except for NH$_4^+$-N accumulation. However, a significantly ($p \leq 0.05$) higher accumulation of NH$_4^+$-N occurred under fertilization −L than its corresponding fertilization +L. Compared with other treatments, NPKC$_R$+L treatment increased the soil contents of AK, P$_{Olsen}$, NO$_3^-$-N, TN, and C:N ratio, with a significant increase recorded for the C:N ratio. The highest accumulation of NH$_4^+$-N occurred under N−L. Liming had strong effects on soil pH, where all the N-based fertilizer applications under +L resulted in a significantly ($p \leq 0.05$) higher soil pH, compared with their equivalents under −L. The highest soil pH of 5.8 was observed under C treatment. This was followed by a soil pH value of 5.3 reported from NPKC$_R$+L, and NPK+L, which was in statistical parity with the soil pH value of 5.2 reported from NP+L treatment (Table 2).

### 3.1.1. Dynamics of Soil pH Change

The current research results revealed that soil pH changes in 0–20 cm and 20–40 cm occurred under different N-based fertilization treatments, −L and +L, during the experimental years (Figure 1). The dynamics of the soil pH changes varied significantly ($p \leq 0.05$) among fertilizations and L treatments. In the −L, the soil pH showed a decreasing trend pattern under all fertilizer treatments, ranging from 5.1 (NPKC$_R$) to 3.8 (NK) in 0–20 cm depth and from 4.9 (NPKC$_R$) to 3.8 (N) in 20–40 cm depth. In contrast, soil pH values showed an increasing trend after L, ranging from 4.5 (N) to 5.5 (NPKC$_R$) and from 4.2 (N and NK) to 5.1 (NPKC$_R$) in 0–20 cm and 20–40 cm, respectively. In spite of the soil's depth and L, soil pH values under all fertilizer treatments were lower, compared with the control treatment. The soil pH values under P-containing fertilizers (NP, NPK, and NPKC$_R$) showed a slightly decreasing trend while exhibiting a considerable decrease under P-omitted fertilizers (N and NK), in −L treatments. On the other hand, higher rising patterns of soil pH were observed due to +L under NPKC$_R$, NPK, and NP, which depicted a pronounced increase from NPKC$_R$ and NPK.

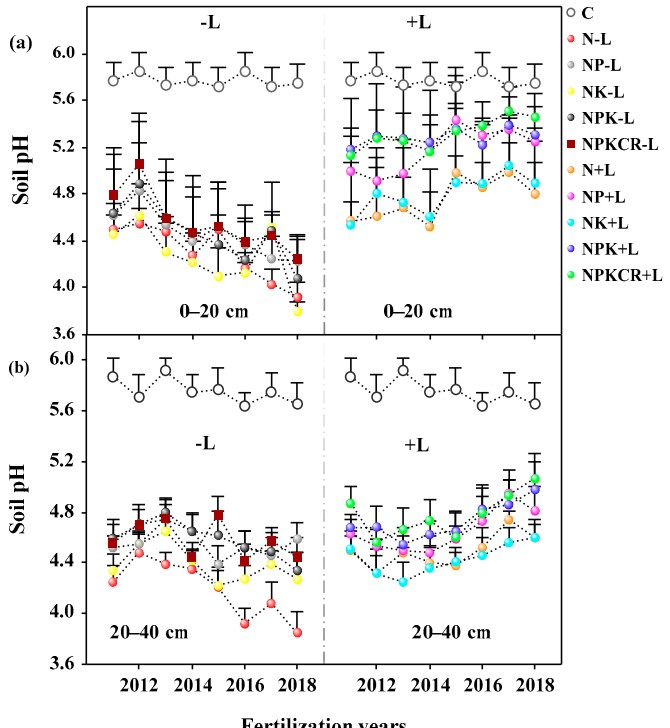

**Figure 1.** (**a**) Trends in soil pH in 0–20 cm; and (**b**) 20–40 cm without L (−L) and with L (+L) fertilizations in 8 years experiment of wheat and maize system. Bars represents SD of means (*n* = 4). C: no fertilizer application or control, N: nitrogen, NP: N and phosphorus (P), NK: N and potassium (K), NPK: N, P, and K, NPKCR: NPK plus crop residue (C$_R$), −L: without lime application, +L: with lime application.

The differences between the soil pH values in the control treatment and other fertilizer treatments were generally much larger in −L, and smaller in +L; irrespective of the depth, the soil samples were taken (Figure 2). Compared with −L, the +L increased the average soil pH values of all fertilizer treatments by 3.0–22.6% in 0–20 cm and 4.2–11.9% in 20–40 cm from 2011 to 2018. This indicates that the dynamic changes in soil pH values between fertilizer treatments under −L and +L were relatively larger in the 0–20 cm depth than those in 20–40 cm depth.

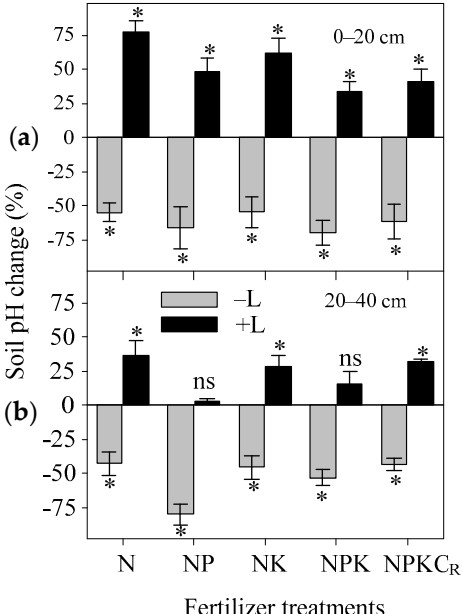

**Figure 2.** (**a**) Soil pH changes (%) between 2011 and 2018 for each fertilization treatment in −L and +L under 0–20 cm; and (**b**) under 20–40 cm soil depths, compared with soil pH in control (C) plot. Bars represent SD of means (*n* = 4); *, denotes significantly increased or decreased soil pH of fertilizer treatments compared with control at $p \leq 0.05$ with ns denoting $p > 0.05$ by LSD test. −L: without lime application, +L: with lime application.

3.1.2. Relationship between Soil pH, Exchangeable Soil Cations, and Years after Liming

For the −L and +L treatments soil pH continued to decline and increase, respectively, over the period of 2011–2018 (Figure 1a,b). Hence, to examine the changes in soil cations in relationship to changes in pH observed over the experiment period we presented the observed pH trend on the x-axis or from high pH to low pH for −L, and the reverse for +L treatment.

The increasing ($p < 0.001$; $r^2 = 0.67$) and decreasing ($p < 0.001$; $r^2 = 0.57$) soil exchangeable $Al^{3+}$ in −L and +L, during the period from 2011 to 2018, had negative relationships with the soil pH (Figure 3a,b). Conversely, the soil pH had a positive relationships of $p < 0.001$; $r^2 = 0.39$ and $p < 0.05$; $r^2 = 0.16$ and with the soil exchangeable $Ca^{2+}$ in −L and +L soils, respectively (Figure 3c,d). The linear regression analysis exhibited that the soil pH had a significant positive relationship ($p < 0.05$; $r^2 = 0.11$) with soil exchangeable $Mg^{2+}$ in +L soil (Figure 3f); while, it had no significant relationship with soil exchangeable $Mg^{2+}$ in −L soil during the experimental period (Figure 3e). As the years after L increased, soil pH also increased strongly ($p < 0.001$; $r^2 = 0.79$) in +L soil (Figure S2a). Similarly, the years after L had a strong positive relationship ($p < 0.001$; $r^2 = 0.68$) with the soil exchangeable $Ca^{2+}$, while it had a weak positive relationship with soil exchangeable $Mg^{2+}$. In contrast, a significant negative relationship ($p < 0.001$; $r^2 = 0.55$) was found between soil exchangeable $Al^{3+}$ and the years after L (Figure S2b).

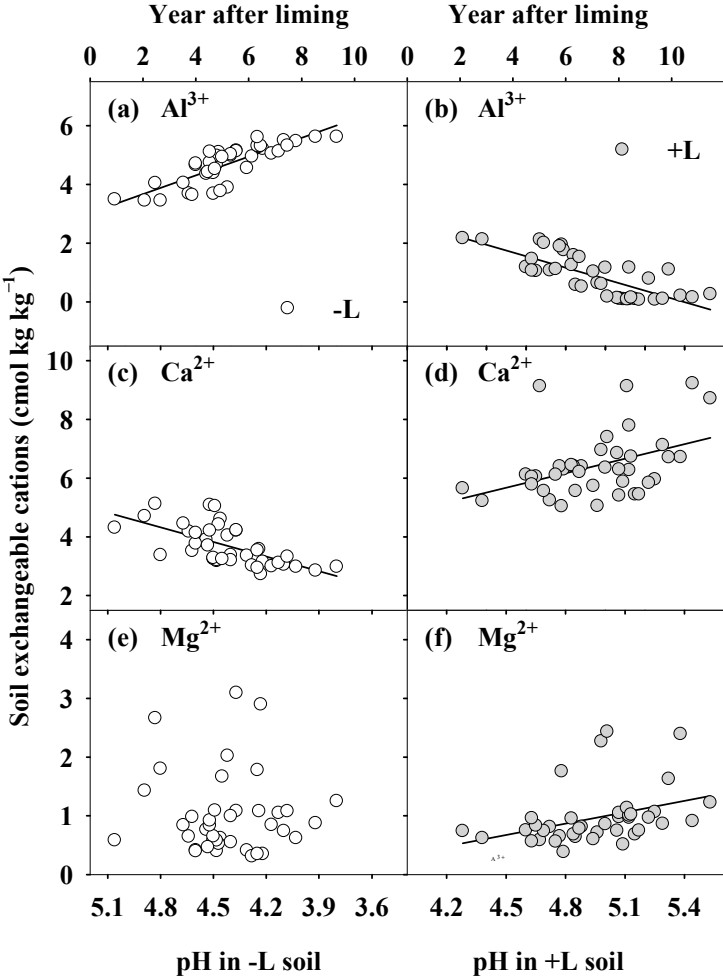

**Figure 3.** (**a**) Relationship between soil pH and soil exchangeable $Al^{3+}$ in $-L$ ($Al^{3+}$ $-L = -2 \times pH + 14.10$, $r^2 = 0.67$); (**b**) in $+L$ ($Al^{3+}$ $+L = -1.95 \times pH + 10.54$, $r^2 = 0.57$); (**c**) soil exchangeable $Ca^{2+}$ in $-L$ ($Ca^{2+}$ $-L = 1.66 \times pH - 3.66$, $r^2 = 0.39$); (**d**) in $+L$ ($Ca^{2+}$ $+L = 1.65 \times pH - 1.74$, $r^2 = 0.16$); (**e**) soil exchangeable $Mg^{2+}$ in $-L$ ($Mg^{2+}$ $-L$ = ns); and (**f**) in $+L$ ($Mg^{2+}$ $+L = 0.64 \times pH - 2.23$, $r^2 = 0.11$). The soil sampling depth was 0–20 cm with ns denoting $p > 0.05$. $-L$: without lime application, $+L$: with lime application.

### 3.2. Wheat and Maize Grain Yields (GY)

The GY of wheat and maize were significantly ($p \leq 0.05$) influenced by a 20-year fertilization, during both the before L (1991–2010) and after L (2011–2018) periods. The average GY of the crops, before L, ranged from 445 kg ha$^{-1}$ (C) to 2002 kg ha$^{-1}$ (NPK) for wheat (Figure 4a) and from 675 kg ha$^{-1}$ (C) and 3305 kg ha$^{-1}$ (NPKC$_R$) for maize (Figure 4b).

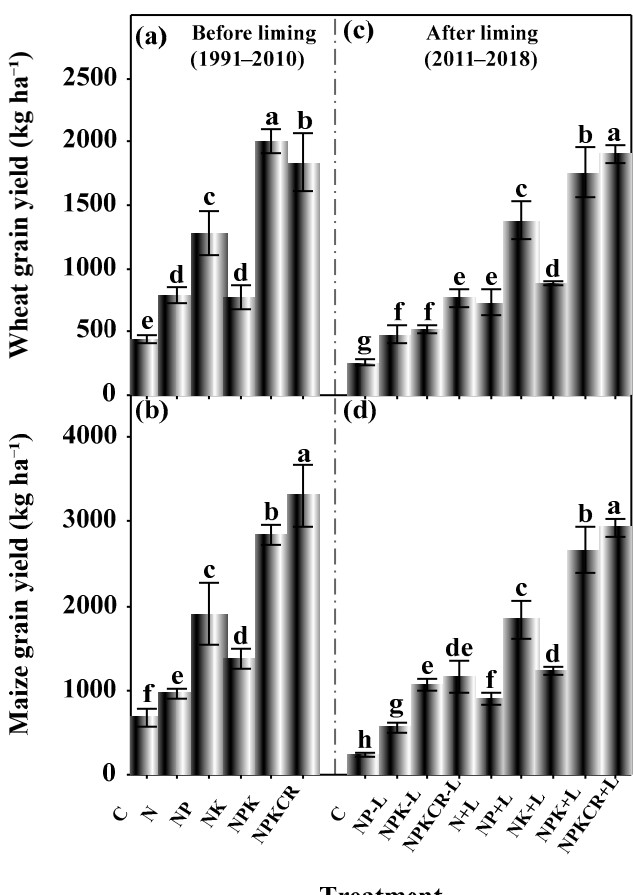

**Figure 4.** (**a**) Average annual wheat; and (**b**) maize grain yields, before liming; (**c**) annual wheat; and (**d**) maize grain yields, after liming, under various fertilization treatments of a field experiment in a wheat-maize rotation. Bars represent SD of means (*n* = 4). Bars with different letters indicate a significant difference at $p \leq 0.05$ by the LSD test. C: no fertilizer application or control, N: nitrogen, NP: N and phosphorus (P), NK: N and potassium (K), NPK: N, P, and K, NPKCR: NPK plus crop residue (CR), −L: without lime application, +L: with lime application.

After L, the mean GY of wheat and maize from NP, NPK and NPKC$_R$ treatments in −L were less than 1000 kg ha$^{-1}$ and 1500 kg ha$^{-1}$, respectively (Figure 4c). However, the GY was not detected from the N and NK application in −L for both crops because of severe soil acidification under these treatments during the period from 2011 to 2018 (Figure 1). In +L, the highest GY (1904 kg ha$^{-1}$) was recorded for wheat and 2928 kg ha$^{-1}$ for maize under NPKC$_R$+L. There was no significant (*p* > 0.05) difference between the GY of wheat under the treatments (NP−L and NPK−L) and (NPKC$_R$+L and N+L) (Figure 4c), and between the GY of maize under the treatments (NPK−L and NPKC$_R$−L) and (NPKC$_R$−L and NK+L) (Figure 4d), during the period 2011–2018.

Moreover, the means GY of wheat obtained from NPKC$_R$, NPK, and NP treatments in +L exceeded the wheat GY obtained from their corresponding treatments in −L by 148%, 236%, and 191%, respectively. Similarly, the maize GY obtained in response to NPKC$_R$, NPK, and NP in +L exceeded, by 150%, 151%, and 227%, respectively, the maize GY obtained from NPKC$_R$, NPK, and NP in −L. The largest −L to +L variation in the GY of wheat (1236 kg ha$^{-1}$) and maize (1759 kg ha$^{-1}$) occurred for NPK and NPKC$_R$, respectively. Furthermore, in an 8-year +L, the P-containing fertilizers (NP, NPK, and NPKC$_R$) remarkably improved the wheat and maize yields compared with the P-omitted (N and NK) and C treatments.

### 3.3. Correlation between Soil Chemical Properties, GY, and NUE Indices

According to the redundancy analysis (RDA), the fertilizer treatments +L such as NP+L, NPK+L and NPKC$_R$+L are grouped separately to the right of principal component 1 (PC1) (Figure 5). In contrast, fertilizations −L, such as N−L, NP−L, and NK−L grouped to the left of PC1, in the loading plot. The rest of the fertilizer treatments are in between these two groups. The first RDA represents 59% of the variation; the second axis contributed 19% of the total variance.

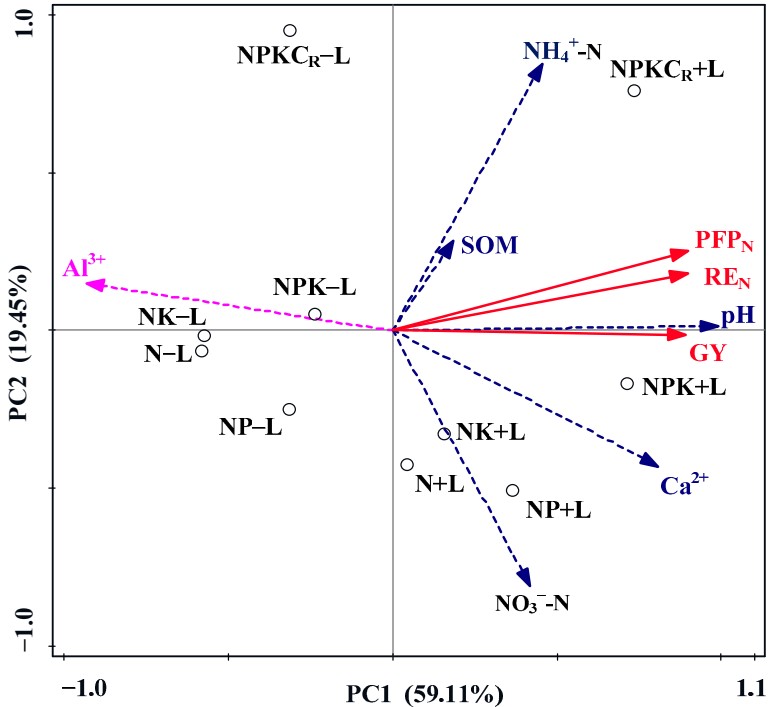

**Figure 5.** CANOCO redundancy analysis scores and loading biplot showing clustering of fertilizers under −L and +L and soil properties (pH, NO$_3^-$-N, NH$_4^+$-N, SOM, Al$^{3+}$, and Ca$^{2+}$) as explanatory variables to explain the variation in the response variables crop N recovery efficiency (RE$_N$), N partial factor productivity (PFP$_N$) and grain yield (GY). The angle between the arrows in the biplot represents the correlation between the corresponding variables with 0 and 180° indicating the maximum positive and negative correlation and 90° indicating no correlation. Soil properties variables are indicated with dashed arrows and NUE and GY variables with solid arrows. N: Nitrogen, NP: N and phosphorus (P), NK: N and potassium (K), NPK: N, P, and K, NPKC$_R$: NPK plus crop residue (C$_R$), −L: without lime application, +L: with lime application.

The NO$_3^-$-N, Ca$^{2+}$, pH, SOM, and NH$_4^+$-N showed positive correlations with GY, RE$_N$, and PFP$_N$ of the crops, where the soil pH predicted the strongest positive correlation with these parameters. Similarly, Al$^{3+}$ correlated negatively with GY, RE$_N$, and PFP$_N$. Of the different soil properties, the soil pH showed the strongest negative correlation with Al$^{3+}$. In addition, crops GY, RE$_N$, and PFP$_N$ were strongly correlated with each other.

### 3.4. Crop N Content and N Uptake

The average grain N content for wheat was generally higher than the value for maize (Table 3). In comparison, the straw N content for wheat was slightly lower than that of maize, regardless of the experimental period and L. Before L, there was slight variation among treatments in grain N content for wheat, straw N contents of wheat and maize, and insignificant difference in grain N content of maize. After L, the N contents were found to be significantly ($p \leq 0.05$) higher when lime was applied in combination with P-containing fertilizers (NP+L, NPK+L, and NPKC$_R$+L) than P-omitted (i.e., N+L and NK+L) fertilizers for grain N content of wheat and maize straw N content (Table 4). Overall, the crops' N

content under all fertilizer treatments decreased in −L, while it increased in +L compared with their corresponding N contents before L.

**Table 3.** Average N content and uptake at maturity in response to different fertilization treatments before liming (over 1991–2010).

| Item | Crop | Plant Part | Treatment | | | | | | LSD ($p \leq 0.05$) |
|------|------|------------|---|---|---|---|---|---|---|
| | | | C | N | NP | NK | NPK | NPKC$_R$ | |
| N content (%) | Wheat | Grain | 1.07 d | 1.63 c | 1.98 ab | 1.90 b | 1.95 ab | 2.12 a | 0.19 |
| | | Straw | 0.45 c | 0.44 c | 0.60 b | 0.67 a | 0.64 ab | 0.63 ab | 0.06 |
| | Maize | Grain | 1.10 a | 1.00 a | 1.22 a | 1.08 a | 1.15 a | 1.14 a | ns |
| | | Straw | 0.87 a | 0.49 b | 1.02 a | 0.92 a | 0.99 a | 0.98 a | 0.31 |
| N uptake (kg ha$^{-1}$) | Wheat | Grain | 4.76 d | 15.04 c | 28.01 b | 17.33 c | 40.32 a | 39.54 a | 6.11 |
| | | Straw | 3.54 f | 9.38 e | 16.88 c | 12.30 d | 26.82 b | 30.38 a | 2.18 |
| | | Total | 8.30 d | 24.91 cd | 44.90 b | 29.63 c | 67.14 a | 69.93 a | 8.89 |
| | Maize | Grain | 7.42 f | 11.11 e | 25.56 c | 17.09 d | 33.79 b | 38.25 a | 2.30 |
| | | Straw | 9.53 c | 8.87 c | 20.65 b | 13.73 c | 24.60 ab | 27.55 a | 6.09 |
| | | Total | 16.95 e | 19.98 e | 46.21 c | 30.82 d | 58.39 b | 65.80 a | 5.87 |

Means followed by different letters within the same row are significantly different from each other at $p \leq 0.05$ by LSD test with ns denoting $p > 0.05$. C: no fertilizer application or control, N: nitrogen, NP: N and phosphorus (P), NK: N and potassium (K), NPK: N, P, and K, NPKC$_R$: NPK plus crop residue (C$_R$).

**Table 4.** Average N content and uptake at maturity in response to different fertilization treatments after liming (over 2011–2018).

| Item | Crop | Plant Part | Treatment | | | | | | | LSD ($p \leq 0.05$) |
|------|------|------------|---|---|---|---|---|---|---|---|
| | | | Lime | C | N | NP | NK | NPK | NPKC$_R$ | |
| N content (%) | Wheat | Grain | −L | 1.01 f | * | 1.69 d | * | 1.76 cd | 1.89 bc | 0.19 |
| | | | +L | - | 1.49 e | 2.72 a | 1.46 e | 2.79 a | 2.90 a | |
| | | Straw | −L | 0.29 f | * | 0.37 e | * | 0.45 d | 0.43 d | 0.05 |
| | | | +L | - | 0.61 ab | 0.65 a | 0.58 b | 0.63 ab | 0.64 a | |
| | Maize | Grain | −L | 0.90 f | * | 0.87 f | * | 0.84 f | 0.86 f | 0.10 |
| | | | +L | - | 1.16 de | 1.37 b | 1.25 cd | 1.35 bc | 1.48 a | |
| | | Straw | −L | 0.44 d | * | 0.73 c | * | 0.78 c | 0.80 c | 0.09 |
| | | | +L | - | 0.89 b | 1.17 a | 0.93 b | 1.15 a | 1.19 a | |
| N uptake (kg ha$^{-1}$) | Wheat | Grain | −L | 2.67 h | * | 8.10 g | * | 9.36 fg | 14.63 e | 5.12 |
| | | | +L | - | 11.53 efg | 39.43 c | 13.38 ef | 51.88 b | 57.07 a | |
| | | Straw | −L | 2.34 i | * | 3.75 h | * | 9.07 ef | 8.70 f | 1.12 |
| | | | +L | - | 7.57 g | 14.30 c | 9.96 de | 17.63 b | 18.88 a | |
| | | Total | −L | 5.01 g | * | 11.85 f | * | 18.43 e | 23.33 e | 6.21 |
| | | | +L | - | 19.10 e | 53.72 c | 23.34 e | 69.50 b | 75.95 a | |
| | Maize | Grain | −L | 2.16 f | * | 4.96 f | * | 9.25 e | 10.14 e | 2.99 |
| | | | +L | - | 11.68 e | 26.11 c | 16.40 d | 37.72 b | 43.60 a | |
| | | Straw | −L | 4.91 j | * | 7.13 f | * | 10.25 e | 11.63 d | 1.08 |
| | | | +L | - | 13.63 c | 16.13 b | 15.13 b | 19.25 a | 19.75 a | |
| | | Total | −L | 7.07 h | * | 12.09 g | * | 19.50 f | 21.77 ef | 4.06 |
| | | | +L | - | 25.31 e | 42.23 c | 31.52 d | 56.72 b | 63.35 a | |

Means followed by different letters within the same row are significantly different from each other at $p \leq 0.05$ by LSD test. C: no fertilizer application or control, N: nitrogen, NP: N and phosphorus (P), NK: N and potassium (K), NPK: N, P, and K, NPKC$_R$: NPK plus crop residue (C$_R$), −L: without lime application, +L: with lime application, *: no data available.

The results also showed that 20-year fertilizations before L, and an 8-year xo-application of lime and fertilizers after L significantly affected the grain, straw, and the total N uptakes

of wheat and maize. Maize took up more N than wheat. The total aboveground N uptake for NP, NPK, and NPKC$_R$ in +L were higher by 353%, 277%, and 225%, respectively, than their corresponding treatments in −L for wheat, while they were higher by 249%, 190%, and 191%, respectively, for maize. The total aboveground N uptakes of both crops for NP, NPK, and NPKC$_R$ after L were generally lower in −L and higher in +L than their equivalents before L (Table 4).

### 3.5. Nitrogen Use Efficiency

The NUE indices (RE$_N$ and PFP$_N$) of both wheat and maize under all fertilizer treatments exhibited similar patterns throughout the study periods, i.e., they generally showed a decreasing trend over 1991–2010 (before L), and 2011–2018 in −L, while showing an increasing trend in +L (Figure 6a–c). Further, the data showed that the variation in RE$_N$ and PFP$_N$ of wheat and maize differed significantly ($p \leq 0.05$) vis-à-vis L, fertilization years, and fertilizer treatments. Before L, the RE$_N$ and PFP$_N$ of both crops decreased significantly after the first few years and then reduced slightly under all treatments. In addition, N-based fertilization −L decreased both the RE$_N$ and PFP$_N$ of wheat and maize during 2011–2018, while it exhibited a steady increase in RE$_N$ and PFP$_N$ when amended with L, in spite of some fluctuations in some years. NPKC$_R$, NPK, and NP showed an apparent superiority over N and NK fertilizers in improving crop NUE, irrespective of L and fertilization years for both crops. Meanwhile, our data showed that no RE$_N$ and PFP$_N$ were detected from P-omitted (N and NK) treatments during 2011–2018 in −L.

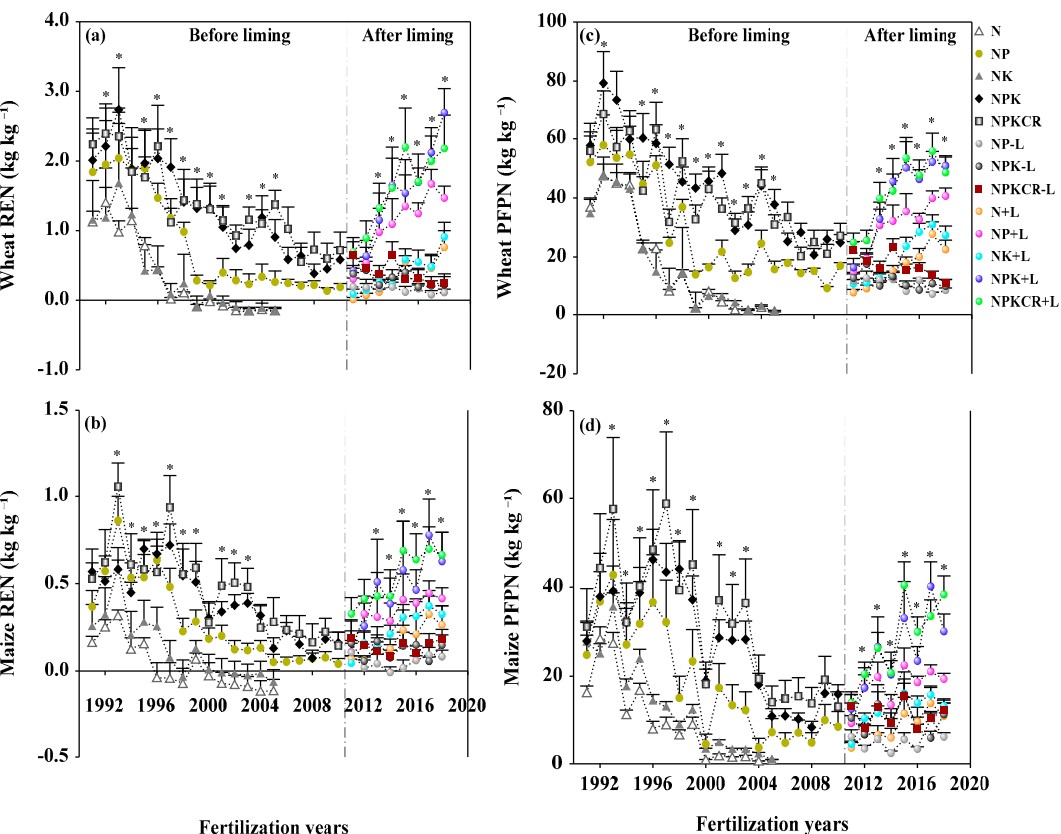

**Figure 6.** (**a**) Variation in N recovery efficiency (REN) of wheat; and (**b**) maize; (**c**) N partial factor productivity (PFPN) of wheat; and (**d**) maize, before and after L. Bars represents SD of means (*n* = 4); *, denotes significant difference among the treatments at $p \leq 0.05$ according to LSD test. N: nitrogen, NP: N and phosphorus (P), NK: N and potassium (K), NPK: N, P, and K, NPKC$_R$: NPK plus crop residue (CR), −L: without lime application, +L: with lime application.

NPKC$_R$−L caused the averaged wheat RE$_N$ and PFP$_N$ values to decrease by 236% and 140%, while NPKC$_R$+L increased them by 15% and 4%, respectively. For maize, NPKC$_R$

application decreased the average value of $RE_N$ by 232% in $-L$ and decreased by 15% in $+L$; however, its application decreased the average value of maize $PFP_N$ by 183% and 13% in $-L$ and $+L$, respectively, compared with its equivalent before L.

## 4. Discussion

### 4.1. Soil Chemical Properties

Nitrogen-based fertilization induced soil acidification is one of the most important factors, adversely affecting several chemical and biological processes in the soils [10,53], which can result in decreased crop production [43]. As indicated in this study, 8 years of lime application significantly ameliorated soil exchangeable $Al^{3+}$ and increased base cations ($Ca^{2+}$ and $Mg^{2+}$), soil pH, and improved the soil nutrient status ($NO_3^--N$, TN, $P_{Olsen}$, and SOM). Lime can neutralize excessive acidic ions (e.g., $Al^{3+}$ and $H^+$), while supplying basic cations (e.g., $Ca^{2+}$ and $Mg^{2+}$) in the soil [17]. The effect of L was found to be better than that of incorporating straw into the soil, which agrees with the previous reports from the same site by several researchers [43,53]. For instance, [53] indicated that 7 years of L significantly increased soil pH under NPK and $NPKC_R$, compared with the soil pH in $-L$. This is due to lime application increasing soil pH in acidic soil because of its strong capacity in acid neutralization [54]. In contrast to our findings, [55] showed from an incubation experiment that soil pH was more significantly increased under $NPKC_R$ than under NPK, regardless of L. This might be associated with the amount of organic materials applied in the field experiment, which perhaps was lower than what was added in the incubation experiment [54].

The application of the NPK fertilizer with recycling of $C_R$ resulted in a higher available K (AK) content in the 0–20 cm soil layer than when no K fertilizer was applied (Table 1). This result is in line with a previously reported study that the further application of K into the soil from $NPKC_R$ through straw incorporation can remarkably increase AK compared with the NPK alone application in a 26-year field experiment [42]. Therefore, in the present study, the $NPKC_R+L$ increased AK, compared with the other treatments (Table 2).

Compared with other treatments, N application significantly increased the accumulation of $NH_4^+-N$, and decreased $NO_3^--N$ and SOM in $-L$ soil, possibly due to lower soil pH in $-L$ decreased nitrification, which thereby increased $NH_4^+-N$ and decreased $NO_3^--N$ accumulations. Relatedly, mineralization of organic N [55] and nitrifying bacteria that convert $NH_4^+$ to $NO_3^-$ [56], were generally performed poorly under low soil pH. Therefore, SOM and TN contents in this experiment were lower under $-L$ than $+L$ (Table 2). Lime promotes the mineralization of SOM, and consequently, it increases the mobilization of plant nutrients from the organic matter and also decreases the content of organic carbon [57].

Thus, variations in soil chemical properties due to the addition of $C_R$ were higher under $+L$ than those under $-L$ (Table 2), which agrees with the previous results [42]. In this study, the effect of L on the soil pH was more significant for $NPKC_R$ treatment than other treatments under 0–20 cm (Figure 1). This supports the assumption that returning plant residues to the soil and applying chemical nutrients may have more impact on acidic surface soil compared with its application to subsurface acidic soil [58]. Thus, 8 years of $+L$ between 2011 and 2018, may have promoted soil pH increases under N and NK (P-omitted) treatments more than others in both soil depths (pH change rate of 75% in 0–20 cm and 35% in 20–40 cm depth). This might be attributed to the severe soil acidity observed under N and NK applications compared with the NP, NPK, and $NPKC_R$ (P-containing) fertilizations. Consistent with this argument, soil acidification from long-term use of inorganic N fertilizer can be considerably reduced by $\geq$5-year [16] and 7-year [43] lime amendment.

The soil P availability is very sensitive to soil pH [59]. Therefore, in our experiment, $P_{Olsen}$ was significantly ($p \leq 0.05$) affected by L (Table 2). The highest value of $P_{Olsen}$ observed under P-containing fertilizer compared with P-omitted treatments was expected due to more input of P into the soil from NP, NPK, and $NPKCR$ fertilizers. The positive effect of $C_R$ incorporation [60] and L on P availability through decreasing P fixation with



oxides of Fe and Al in highly acidic soils was reported in previous studies [61]. This is in line with our results that the highest $P_{Olsen}$ was observed in $NPKC_R+L$ treatment. Relatedly, [62] reported that lime application significantly increased the $P_{Olsen}$ in acidic soil.

The highest soil C:N ratio in the control treatment compared with N fertilization might be due to the low supply of N fertilizer that leads to a higher soil C:N ratio. However, the highest soil C:N ratio observed under $NPKC_R$, might be due to the relatively higher carbon input through $C_R$ incorporation (Table 2). Except for $NPKC_R$, the soil C:N ratio of all treatments relatively decreased during 2011–2018, compared with their corresponding ratio during 1991–2010. The decreased C:N ratio under chemical N-based fertilization can be due to low and high supply of carbon and nitrogen, respectively. In contrast, [63] the reported a stable soil C:N ratio for the $NPKC_R$ treatment in the long-term fertilization might be due to the linear relationship between SOC and TN contents [64]. The soil C:N ratio plays a significant role in N mineralization and microbial activities [65]. Immobilization due to the high C:N ratio is a major mechanism by which $C_R$ can ameliorate soil acidity [66], and this can reduce the risk of nitrate leaching.

### 4.2. Grain Yield, N Content, N Uptake, and N Use Efficiency

In the previous [11] and present studies, it was reported that soil acidification decreased soil pH through decreasing nitrification; thus, leading to lower $NO_3^-$-N accumulation. This situation adversely affected wheat and maize GY (Figure 4); N uptake (Table 4); $RE_N$ (Figure 6a,b) and $PFP_N$ (Figure 6c,d), in this study. Most of the previous studies at this site focused on the influence of long-term fertilization on soil properties, crop yield, and nutrients availability and use efficiencies, where the results indicated that long-term N-based fertilizations −L negatively influenced these parameters [8,9,39]. This may be ascribed to the fact that long-term inorganic N fertilization −L induces soil acidification [67].

However, previous reports from greater than 5-years L had a positive influence on crop yields [23,62,68], N-uptake [69], and P-use efficiency [43], because of its various positive effects on the physical, chemical, and biological properties of the soil. Inconsistent with the present study (Figures 3–6), [43] noted that L can significantly reduce soil acidification by decreasing soil exchangeable acidic cation ($Al^{3+}$) and by increasing soil exchangeable base cations ($Ca^{2+}$), which, in turn, increases crop productivity, nutrient uptake and use efficiency.

In addition, L enhances the root structure and growth of plants. This has a beneficial effect on the uptake of nutrients [70]. That is likely why, compared with −L, +L significantly increased GY, N contents, N uptakes, $RE_N$, and $PFP_N$ in the present study. The significantly drastic reductions in wheat and maize yields in response to N and NK compared with other treatments in this study might be due to the further occurrence of soil acidification, increased $Al^{3+}$, and/or lower nutrient availability under these treatments.

The continuous use of N alone, for 42 years, had the most acidifying effect with the soil pH value declining from 5.8 to 4.5 and adversely affects wheat and maize GY [69]. The higher GY under NPK and $NPKC_R+L$ in this study might be attributed to the higher soil $pH_{H20}$, $NO_3^-$-N, SOM, $P_{Olsen}$, AK and TN contents and N uptake observed under these treatments than other treatments.

Because of the accumulation of SOM over time, the application of $NPKC_R$ may have improved the physical and biological environment of the soil and also provided an adequate supply of nutrients, favoring crop growth [71,72]. Relatedly, in our study, the application of $NPKC_R+L$ was more useful to maintain a high yield than other treatments. The aboveground (grain plus straw) N concentrations in this study are also in line with the previous reports [73], that the N accumulation for the whole wheat and maize was 20.1–29.4 g kg$^{-1}$, and 6.2–22.8 g kg$^{-1}$, respectively.

N uptake may be negatively affected by lower soil pH via primarily affecting the processes that make nitrogen available for crops, i.e., mineralization and nitrification [29]. Compared with other treatments, $NPKC_R+L$ significantly increased the N uptake for maize. However, there was no significant difference exhibited in $NPKC_R$ for wheat than the NPK,

showing that the organic fertilizer amendment under long-term L can bring more N to the next cropping season from returned $C_R$ more for maize than for wheat. In addition, since maize has a higher N requirement than wheat [74], straw incorporation of crops promoting soil N availability might benefit maize more than wheat. In response to fertilizations before L, the drastic decline in the $RE_N$ and $PEP_N$ of wheat and maize after the first few years and then the slight reduction under all other treatments can be due to the phenomena where the soil pH, GY, and N uptake were reduced, in that order (Figures 1 and 4, Tables 3 and 4). Consistent with the results of our study, other researchers [36] also indicated that long-term application of N-based fertilizations resulted in a very low 16–18% average for on farm $RE_N$ and a $PFP_N$ of 30–37 kg kg$^{-1}$ for the intensive wheat–maize system. The lowest or negative values or no-$RE_N$ and no-$PFP_N$ observed under N and NK (Figure 6) were because of their meager or no GY relative to the control and other fertilizer treatments.

Likewise, the steadily rising trends in both indices of wheat and maize when lime and fertilizers are applied during the study years confirmed our hypothesis and agreed with the results of the previous report that $RE_N$ and $PFP_N$ can be significantly improved in a 22-year L [75]. This might be associated with the fact that liming duration had a significant effect on the soil exchangeable cations (Figure S2b) and soil pH (Figure S2a). Therefore, in our study, soil exchangeable cations were positively influenced by the duration of L, indicating that increasing L years significantly increased exchangeable $Ca^{2+}$ and soil pH, while it decreased soil exchangeable $Al^{3+}$, which can in turn, enhanced crop yield and NUE. The highest increasing of $RE_N$ and $PFP_N$ of crops observed from $NPKC_R$ and NPK under +L can be attributed to the higher soil pH and yield improvements.

Here, +L increased the soil pH and its effect on NUE by improving N mineralization and its availability to crops, thus encouraged the crops to increase NUE in response to the fertilizer application [29]. Regardless of the L effect, the co-application of N and P resulted in a significant improvement in $RE_N$ (by more than 40%) compared with the N alone or with the NK application. Thus, our present results suggest that N alone or with K application had no marked effect on $RE_N$ and $PFP_N$ throughout the experimental period. The adequate K availability to support N use occurred more when K interacted with P than with N [76], and decreased P accumulation which can limit plant growth [77]. Nevertheless, further studies should explain how N and K interact to support crop production in lime amended acidic soil. RDA also showed that, of different soil chemical properties, soil pH was the most important factor that affected GY, $RE_N$ and $PFP_N$, in an 8-year fertilization under +L (Figure 5). This may have occurred, due to the +L increased $Ca^{2+}$, $Mg^{2+}$ (Figure 3d,f), and reduced the impact of Al toxicity [22,78,79] (Figure 3b). The finding is also in line with the previous reports [14,15] that soil pH directly or indirectly affects the soil's biochemical properties, thereby impacting plant growth.

## 5. Conclusions

In our study, soil chemical properties, wheat, and maize yields, and NUE significantly deceased in long-term N-based fertilization under −L. However, 8 years of L significantly improved wheat and maize crop yields and NUE through increasing soil pH, exchangeable base cations such as $Ca^{2+}$, $Mg^{2+}$, and alleviating the toxicity of $Al^{3+}$, with soil pH having the most pronounced influence. The results also showed that both yield and NUE were significantly higher under NPK+L and $NPKC_R$+L than NPK−L and $NPKC_R$−L. This might be related to the retention of SOM through $C_R$ incorporation and +L induced soil pH effects on soil available N and N uptake by the crops. Therefore, based on our study, we suggest that 8 years of applications of combined chemical fertilizer (NPK), $C_R$, and +L (i.e., $NPKC_R$+L) can effectively contribute to the improvement of crop yields and NUE in acidic soils. Although not evaluated in this study, the extant studies have shown that soil $NO_3{}^-$-N carryover can affect wheat and maize responses to applied N [80,81]. Hence, future research should include soil $NO_3{}^-$-N carryover in each crop growing season to meet the crop's nutrient requirement level and maximize NUE at a given yield level in acidic soil under a wheat-maize rotation.

**Supplementary Materials:** The following are available online at https://www.mdpi.com/article/10.3390/agronomy11102069/s1. Figure S1. Experimental layout and design. Figure S2. (a) Relationship between year after liming and soil pH (soil pH = 0.07x + 4.62, $r^2$ = 0.79); and (b) soil exchangeable cations ($Al^{3+}$ = −0.14x + 1.50, $r^2$ = 0.54; $Ca^{2+}$ = 0.26x + 5.26, $r^2$ = 0.68; and $Mg^{2+}$ = 0.07x + 0.66, $r^2$ = 0.42). The soil sampling depth was 0–20 cm.

**Author Contributions:** Conceptualization, N.A.D. and D.L.; methodology, N.A.D.; software, D.L. and J.D.; validation, J.H., T.H. and L.Z.; formal analysis, B.W.; investigation, D.L.; resources, H.Z.; J.H. and B.W.; data curation, Y.X. and J.D.; writing—original draft preparation, N.A.D.; writing—review and editing, N.A.D., S.A., M.N.K., T.G.L. and T.H.; visualization, L.L. and S.L.; supervision, H.Z.; project administration, H.Z.; funding acquisition, H.Z. All authors have read and agreed to the published version of the manuscript.

**Funding:** This research was financially supported by the National Natural Science Foundation of China (No. 41671301), the National Key Research and Development Program of China (2016YFD0300901 and 2017YFD0800101), and the Fundamental Research Funds for Central Non-profit Scientific Institution (161032019035, 1610132020021,1610132020022, 1610132020023).

**Data Availability Statement:** Data is contained within the article.

**Acknowledgments:** We are grateful to all the managers of long-term experiments at the National Observation Station of Qiyang Agri-ecology System, Hunan province, China.

**Conflicts of Interest:** The authors declare no conflict of interest in this manuscript.

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
