# Peer review of "Long-Term Fertilization and Lime-Induced Soil pH Changes Affect Nitrogen Use Efficiency and Grain Yields in Acidic Soil under Wheat-Maize Rotation"

_agronomy, doi:10.3390/agronomy11102069_

Round 1
Reviewer 1 Report
I believe the manuscript has improved since the last round of reviews since the authors have made several changes in line with previous reviewers’ comments. The methods used are clearer and the manuscript better presented overall. However, I have identified passages throughout the text that I believe still need attention before the manuscript can be published, as detailed below. The text would also benefit from extensive English editing to improve clarity.
Abstract
The study location should be specified in the Abstract. I’m not convinced you need to refer to soil pH as pHH2O, here and elsewhere. I’d suggest you indicate what method was used to measure soil pH in the Methods section, and then refer to it as just soil pH throughout. The last sentence of the Abstract comes across as a generalisation (L40-42), but do you know this to be true under any type of climate and soil? Or is it applicable to your site conditions specifically? Please rewrite it.
Introduction
The first paragraph is poorly contextualised. Are the statements about fertilizer application applicable globally, or specifically to certain regions? Please clarify. Also, you state ‘widely recognised’ in the first sentence but only provide one reference. I’d suggest you either rephrase this sentence or provide more evidence to support this claim.
In the second paragraph, some sentences come across as overgeneralised (L69-70, L72-73). Are these statements about soil pH and acidification applicable specifically to wheat-maize rotations or to acidic soils under arable agriculture in general? Or in dry climates? This paragraph needs to be better contextualised.
L95-97: is this the case everywhere in the world or only in south China? Please be more specific.
L104-106: please rewrite this sentence for clarity.
L108-109: you have not introduced -L and +L in the main text prior to this point. You should clarify what they mean.
L111: in south China?
Methods
L123: I’d call this a figure rather than a table.
L182-184, L195-197: can you provide references for these methods?
L216: ‘fertilizer x L’ or ‘fertilizer +L’? The bracketed term suggests the latter, or are you referring to their interaction? Were year and the individual effects of fertilizer and liming also fitted as fixed effects, or just their interaction term? This is quite confusing, please clarify.
L231, L240: do you mean to say the effect of time?
Results
L260-261: seems redundant and repeats what was said in the previous sentence.
L261-263: I’m not convinced this is correct. By looking at Table 2, the NPK +L treatment seems to have had a similar effect on soil parameters, but you single out NPKCR +L. Generally, it seems there were no statistical differences between them.
L266-268: wasn’t the highest soil pH under C -L? That’s what Table 2 is showing.
Table 2: the supplementary material suggests you had a control treatment in the limed plots. Why are the results for this treatment not shown in table 2?
Fig. 1: what are the red triangles and grey stars in the -L panels? Are they N and NK? They are not indicated in the figure caption. I wonder if it would be clearer to have a legend next to the figure rather than a list in the figure caption.
L311-312: I do not understand this sentence, it seems to contradict the results shown in Fig. 3a as the relationship in both -L and +L seem to show the same trend. Please check.
L315-318: this also seems wrong. The relationship seems to be similar for both -L and +L. Please revise.
L336-347, L355-360: these paragraphs are badly written and difficult to relate to the figure. You seem to have four different panels in Fig. 4. I’d suggest naming them a to d and referring to each panel after each result is presented in the text (e.g., Fig. 4a, Fig. 4b, etc). At the moment, it’s hard work to read the text and the figure in tandem.
L362: long-term is relative; I’d question whether eight years can be qualified as long term. I’d suggest referring to it as the duration of the experimental period (and what the duration was between brackets).
L370-372: this seems more appropriate for the figure caption than the main text.
L378: might also be worth indicating the different colours the soil and crop properties are presented in the figure (blue and red). I’d suggest also indicating the symbol used to represent the fertilizer treatments in the figure caption.
L396: please specify the number of years instead of using ‘long term’.
L396-402: are these results related to the data presented in Table 4? Please clarify.
Table 4: please indicate what the asterisks mean in the table caption.
L423-424: does this apply to both crops?
L427-428: this seems to repeat what was stated in the previous paragraph.
Figs. 6 and 7: I wonder if it would be better to have these two figures as four panels (a to d) in one figure, with a legend next to it indicating what the symbols mean. In both figures, some of the error bars seem to stand alone and not connected to any symbol.
Discussion
L450: one of the most important factors for what exactly?
L475-477: it is difficult to understand the meaning of this sentence. Please rewrite it.
L487: what changes were these? Was there a reduction or increase in soil pH?
L507-508: please be more specific. What is long term here? State the length of time you are referring to.
L511: I’m not clear what immobilization you are referring to here.
L536: is this a general comment? It sounds very specific to your study. Please rephrase it.
L542: what soil physical and biological properties did you measure to support this? Do you mean to say, ‘the application of NPKCR may have improved…’?
L546-547: is this correct?
L586, L591, L593, L600: I’d suggest you specify what length of time you are referring to instead of using ‘long-term’.
Author Response
Reviewer 1
Point by point response to comments and suggestions raised by Reviewer 1
Manuscript ID: agronomy-1366475.
Manuscript Title: Long-term fertilization and lime-induced soil pH changes affect nitrogen use efficiency and grain yields in acidic soil under wheat-maize rotation.
Authors: Nano Alemu Daba, Li Dongchu, Huang Jing, Han Tianfu, Zhang Lu, Sehrish Ali,
Muhammad Numan Khan, Du Jiangxue, Liu Shujun, Tsegaye Gemechu Legesse, Liu
Lisheng, Xu Yongmei, Zhang Huimin and Wang Boren.
General comment.
Point 1: I believe the manuscript has improved since the last round of reviews since the authors have made several changes in line with previous reviewers’ comments. The methods used are clearer and the manuscript better presented overall. However, I have identified passages throughout the text that I believe still need attention before the manuscript can be published, as detailed below. The text would also benefit from extensive English editing to improve clarity.
Response 1: We are very thankful to the reviewer for the constructive comments and suggestions.
Abstract
Point 2: The study location should be specified in the Abstract. I’m not convinced you need to refer to soil pH as pHH2O, here and elsewhere. I’d suggest you indicate what method was used to measure soil pH in the Methods section, and then refer to it as just soil pH throughout. The last sentence of the Abstract comes across as a generalisation (L40-42), but do you know this to be true under any type of climate and soil? Or is it applicable to your site conditions specifically? Please rewrite it.
Response 2: -We have included the study location, as “South China”. Based on your suggestion, the previously mention “soil pHH2O” is now changed to “soil pH” throughout the paper. The last single statement across L40-42 in particular and last four sentences of the abstract in general was improved.
Introduction
Point 3: The first paragraph is poorly contextualised. Are the statements about fertilizer application applicable globally, or specifically to certain regions? Please clarify. Also, you state ‘widely recognised’ in the first sentence but only provide one reference. I’d suggest you either rephrase this sentence or provide more evidence to support this claim.
Response 3: The statement about fertilizer application is applicable to China; hence we have included “China” in this sentence. Similarly, we have added further references to support the claim ‘widely recognised’ in the first paragraph.
Point 4: In the second paragraph, some sentences come across as overgeneralised (L69-70, L72-73). Are these statements about soil pH and acidification applicable specifically to wheat-maize rotations or to acidic soils under arable agriculture in general? Or in dry climates? This paragraph needs to be better contextualised.
Response 4: They were referring about the Soil acidification effects in dry land system of red soil in south China in general, not specific to wheat-maize rotations. Due to the phrase “wheat-maize rotations” may make confusion in its current context; we have excluded it from this sentence and the paragraph is better contextualised.
Point 5: L95-97: is this the case everywhere in the world or only in south China? Please be more specific.
Response 5: This is in the case of China’s agriculture and this line was the continuation of the previous idea, where we already specified “China’s agriculture”. To be more specific, we have included China in this line as well.
Point 6: L104-106: please rewrite this sentence for clarity.
Response 6: Thanks for your comment. We have modified this line as per your comment.
Point 7: L108-109: you have not introduced -L and +L in the main text prior to this point. You should clarify what they mean.
Response 7: Thanks for the correction. We have clarified -L, as without liming, and +L, as with liming both in abstract and introduction parts.
Point 8: L111: in south China?
Response 8: Thanks. Yes, it is South China, and we have included.
Methods
Point 9: L123: I’d call this a figure rather than a table.
Response 9: Thanks. We named it as Figure S1, based on your suggestion.
Point 10: L182-184, L195-197: can you provide references for these methods?
Response 10: Referring different published literatures having the same parameters with our current parameters, the specific references were not mentioned for these methods. However, we have made some modification on method in L182-184, based on published literature.
Point 11: L216: ‘fertilizer x L’ or ‘fertilizer +L’? The bracketed term suggests the latter, or are you referring to their interaction? Were year and the individual effects of fertilizer and liming also fitted as fixed effects, or just their interaction term? This is quite confusing, please clarify.
Response 11: Thanks for your comment. In our study both fertilizer x L and ‘fertilizer +L’ referring our treatments after liming. However, fertilizer +L is constantly used to refer fertilizer and liming treatments throughout the paper, hence we have replaced ‘fertilizer x L’ with ‘fertilizer + L’ here as well. Fertilizers, liming and fertilizer +L were fitted as fixed effects, while year was fitted as random effect.
Point 12: L231, L240: do you mean to say the effect of time?
Response 12: Yes. However, in our study the time was grouped into two periods. (1) Before liming (1991-2010), which was 20-year totally; (2) After liming (2011- 2018), which was 8-year. Thus, the effect of time in our case was 20 years and 8 years under before liming and after liming, respectively.
Results
Point 13: L260-261: seems redundant and repeats what was said in the previous sentence.
Response 13: Thanks. We have checked the redundant and repeat, and revised this sentence.
Point 14: L261-263: I’m not convinced this is correct. By looking at Table 2, the NPK +L treatment seems to have had a similar effect on soil parameters, but you single out NPKCR +L. Generally, it seems there were no statistical differences between them.
Response 14: Thanks for your comments. We agree that, numerically, the values of mentioned soil properties under NPKCR +L were larger than those under NPK +L, but not statistically. Therefore, considering the comment, we have revised this sentence and others regarding to soil properties.
Point 15: L266-268: wasn’t the highest soil pH under C -L? That’s what Table 2 is showing.
Response 15: Thanks for the correction. Yes, the highest soil pH was observed under control (C) treatment. We have corrected and revised accordingly.
Point 16: Table 2: the supplementary material suggests you had a control treatment in the limed plots. Why are the results for this treatment not shown in table 2?
Response 16: We have the same (one) control (i.e. C) treatment both under -L and +L. We hope the supplementary material is also suggests the same.
Point 17: Fig. 1: what are the red triangles and grey stars in the -L panels? Are they N and NK? They are not indicated in the figure caption. I wonder if it would be clearer to have a legend next to the figure rather than a list in the figure caption.
Response 17: Thanks for the comment. This figure, including its legend, is re-plotted and revised, based on the comment and suggestion.
Point 18: L311-312: I do not understand this sentence, it seems to contradict the results shown in Fig. 3a as the relationship in both -L and +L seem to show the same trend. Please check.
Response 18: Thanks for the comment. We have checked this sentence and it seems confusing in the previous format and style. Here, the relationship between soil pH and exchangeable cations (in figure 3) were based on the sequence of the experimental years after liming. Accordingly, there were two scenarios regarding soil pH. The 1st was, the soil pH got decreasing from 2011 to 2018, under -L soil. In the 2nd scenario the soil pH was increasing across 2011 – 2018, under +L soil. However, it was difficult to understand these situations from the previous Figure 3, as already commented. Therefore, re-plotted this figure having 6 separate panels from a to f, for better understanding.
Point 19: L315-318: this also seems wrong. The relationship seems to be similar for both -L and +L. Please revise.
Response 19: Thanks for the comment. This is the part of response 18 (mentioned above). Therefore, please kindly see our response to point 18.
Point 20: L336-347, L355-360: these paragraphs are badly written and difficult to relate to the figure. You seem to have four different panels in Fig. 4. I’d suggest naming them a to d and referring to each panel after each result is presented in the text (e.g., Fig. 4a, Fig. 4b, etc). At the moment, it’s hard work to read the text and the figure in tandem.
Response 20: Thanks for the comment. Considering comments and suggestions, we have revised these paragraphs and named the panels from a to d. Similarly, we have referred to each panel after each result is present in the text.
Point 21: L362: long-term is relative; I’d question whether eight years can be qualified as long term. I’d suggest referring to it as the duration of the experimental period (and what the duration was between brackets).
Response 21: Thanks for the comment. Considering the comment, we have specified long-term as eight years and revised this sentence.
Point 22: L370-372: this seems more appropriate for the figure caption than the main text.
Response 22: Thanks. Base on the comment and suggestion, we have moved this statement to Figure 5 caption.
Point 23: L378: might also be worth indicating the different colours the soil and crop properties are presented in the figure (blue and red). I’d suggest also indicating the symbol used to represent the fertilizer treatments in the figure caption.
Response 23: Considering the suggestion, we have indicated the symbol used to represent the fertilizer treatments in the figure caption of revised paper
Point 24: L396: please specify the number of years instead of using ‘long term’.
Response 24: We have specified the number of years instead of using ‘long term’ in the revised paper
Point 25: L396-402: are these results related to the data presented in Table 4? Please clarify.
Response 25: We have clarified and the results are from the data presented in Table 4.
Point 26: Table 4: please indicate what the asterisks mean in the table caption.
Response 26: Thanks for the comment. The asterisks stands for “no data available” and we have indicated in the table caption of revised paper.
Point 27: L423-424: does this apply to both crops?
Response 27: Yes. We have revised this sentence
Point 28: L427-428: this seems to repeat what was stated in the previous paragraph.
Response 28: Thanks for the comment. Considering the comment, we have read thoroughly this and previous paragraphs to check whether the idea was repeated or not. The sentence was revised, as the idea was repeated.
Point 29: Figs. 6 and 7: I wonder if it would be better to have these two figures as four panels (a to d) in one figure, with a legend next to it indicating what the symbols mean. In both figures, some of the error bars seem to stand alone and not connected to any symbol.
Response 29: Based on your suggestion, we have merged figures 6 and 7 into a figure having four panels (a to d) with a legend next to it indicating what the symbols mean. Sorry, we have tried to understand the comment about error bars seem to stand alone and not connected to any symbol, but we can’t exactly understand it. However, we need to confirm you that, the error bars were assigned automatically by software, not manually.
Discussion
Point 30: L450: one of the most important factors for what exactly?
Response 30: Thanks for the comment. This was to refer “Nitrogen based fertilizations”. Due to agriculturally induced soil pH is seems general, we have replaced it with “Nitrogen based fertilizations” in the revised paper.
Point 31: L475-477: it is difficult to understand the meaning of this sentence. Please rewrite it.
Response 31: Considering the comment, we have improved and revised this sentence.
Point 32: L487: what changes were these? Was there a reduction or increase in soil pH?
Response 32: These were to refer increases under +L soil. So, we have replaced with increases, and revised this sentence.
Point 33: L507-508: please be more specific. What is long term here? State the length of time you are referring to.
Response 33: Thanks. We have stated as a 28-year instead of long-term
Point 34: L511: I’m not clear what immobilization you are referring to here.
Response 34: Immobilization is the conversion of an element from an inorganic to organic form by microorganisms. In our case, under high C/N ratio, N nutrient might be limiting to microbial metabolism, hence, nutrients liberated by mineralization (from crop residue application) will be taken up by microorganisms for microbial metabolism. This could reduce the risk of nitrate leaching and ameliorate soil acidity.
Point 35: L542: what soil physical and biological properties did you measure to support this? Do you mean to say, ‘the application of NPKCR may have improved…’?
Response 35: Thanks. Based on your suggestion, we have revised this sentence by including …. may have …
Point 36: L546-547: is this correct?
Response 36: Thank you for the correction. This sentence was wrongly written and now corrected and revised.
Point 37: L586, L591, L593, L600: I’d suggest you specify what length of time you are referring to instead of using ‘long-term’.
Response 37: Thank for the comment and suggestion. We have specified the length of time as a 20-year and a 8-year before and after liming, respectively, throughout the paper.

Reviewer 2 Report
I already read and review revised paper and accepted it for publication.
Author Response
Reviewer 2
Response to comments raised by Reviewer 2
Manuscript ID: agronomy-1366475.
Manuscript Title: Long-term fertilization and lime-induced soil pH changes affect nitrogen use efficiency and grain yields in acidic soil under wheat-maize rotation.
Authors: Nano Alemu Daba, Li Dongchu, Huang Jing, Han Tianfu, Zhang Lu, Sehrish Ali,
Muhammad Numan Khan, Du Jiangxue, Liu Shujun, Tsegaye Gemechu Legesse, Liu
Lisheng, Xu Yongmei, Zhang Huimin and Wang Boren.
Comment: I already read and review revised paper and accepted it for publication.
Author’s response: We are very thankful to the reviewer for reviewing and accepting our paper for publication.

Reviewer 3 Report
This study uses 28 years of field data to determine the effects of N fertilizer on crop yield, nitrogen use efficiency, soil pH, and some other soil properties. This is exciting research on an important topic. Overall, it will contribute to the literature on agronomy and crop production system. Despite my support for this effort, I have some comments for the authors.
Past studies have shown that crops get nitrogen from applied and carryover nitrogen from the previous year. The accumulation of carryover N significantly affects crops yield (wheat and maize in this case). Without accounting for carryover residual nitrogen in the soil, crop production may not be efficient (N deficit and leaching/ nitrification due to over-application), and soil properties could be altered if farmers apply the same fertilizers levels in all crop growing seasons. I understand that the authors mentioned N, P, and K were applied @ 150 kg per ha, 120 kg per ha, 120 kg per ha, respectively, based on the initial soil test in 1990; however, authors at least should acknowledge the fact that soil nitrate-nitrogen levels in each crop growing season should be considered to meet the crop nutrient requirement (optimal nitrogen level) somewhere in the text.
The abstract is so long and has too many details. Please make it short and concise.
What is soil pHH20?
Please try to make sentences short. For example, line 19-21 is so confusing.
Line 22-25 can be combined into a sentence.
There are several typos and spacing issues. Please check it make sure you correct it. For example, add space before "Because" in line 137, before "We" in line 135.
The discussion is also very long. Please make it a little bit shorter without loss of information.
Please rephrase lines 600-602. Do you mean eight years of lime application?
Please mention the limitations of this study and suggest a few future works. Limitations could be something I mentioned earlier.
Author Response
Reviewer 3
Point by point response to comments and suggestions raised by Reviewer 3
Manuscript ID: agronomy-1366475.
Manuscript Title: Long-term fertilization and lime-induced soil pH changes affect nitrogen use efficiency and grain yields in acidic soil under wheat-maize rotation.
Authors: Nano Alemu Daba, Li Dongchu, Huang Jing, Han Tianfu, Zhang Lu, Sehrish Ali,
Muhammad Numan Khan, Du Jiangxue, Liu Shujun, Tsegaye Gemechu Legesse, Liu
Lisheng, Xu Yongmei, Zhang Huimin and Wang Boren.
General comment.
Point 1: This study uses 28 years of field data to determine the effects of N fertilizer on crop yield, nitrogen use efficiency, soil pH, and some other soil properties. This is exciting research on an important topic. Overall, it will contribute to the literature on agronomy and crop production system. Despite my support for this effort, I have some comments for the authors.
Response 1: Thank very much for the very positive and constructive comments.
Point 2: Past studies have shown that crops get nitrogen from applied and carryover nitrogen from the previous year. The accumulation of carryover N significantly affects crops yield (wheat and maize in this case). Without accounting for carryover residual nitrogen in the soil, crop production may not be efficient (N deficit and leaching/ nitrification due to over-application), and soil properties could be altered if farmers apply the same fertilizers levels in all crop growing seasons. I understand that the authors mentioned N, P, and K were applied @ 150 kg per ha, 120 kg per ha, 120 kg per ha, respectively, based on the initial soil test in 1990; however, authors at least should acknowledge the fact that soil nitrate-nitrogen levels in each crop growing season should be considered to meet the crop nutrient requirement (optimal nitrogen level) somewhere in the text. Please see the following two pieces of literature.
Response 2: Thank you very much for the comments and providing us supporting literatures. We have read thoroughly the literatures, and understand that soil residual NO3- -N influences crop response to applied N. Therefore, considering the comment and suggestion raised, we acknowledged the aforementioned fact under conclusion part of our current study.
Raun, W. R., Dhillon, J., Aula, L., Eickhoff, E., Weymeyer, G., Figueirdeo, B., ... & Fornah, A. (2019). Unpredictable nature of the environment on nitrogen supply and demand. Agronomy Journal, 111(6), 2786-2791.
Dhakal, C., Lange, K., Parajulee, M. N., & Segarra, E. (2019). Dynamic optimization of nitrogen in plateau cotton yield functions with nitrogen carryover considerations. Journal of Agricultural and Applied Economics, 51(3), 385-401.
Point 3: The abstract is so long and has too many details. Please make it short and concise.
Response 3: Thanks for the comment. We have revised and made short and concise the abstract, based on your comment.
Point 4:What is soil pHH20?
Response 4: This was to refer the pH in the soil solution. Now we have changed the “soil pHH20” into “soil pH” throughout the paper.
Point 5:Please try to make sentences short. For example, line 19-21 is so confusing.
Response 5: Thanks for the comment. We have made short some of the long sentences in the revised the paper.
Point 6:Line 22-25 can be combined into a sentence.
Response 6: We have combined them into a sentence.
Point 7:There are several typos and spacing issues. Please check it make sure you correct it. For example, add space before "Because" in line 137, before "We" in line 135.
Response 7: Thanks for the comment. Considering the comment, we have corrected the types and spacing issue throughout the paper.
Point 8:The discussion is also very long. Please make it a little bit shorter without loss of information.
Response 8: Thanks for the comment. We have systematically reduced the discussion part a little bit, as per your suggestion in the revised paper.
Point 9:Please rephrase lines 600-602. Do you mean eight years of lime application?
Response 9: Thanks. Considering the comment, we have rephrased this sentence in the revised paper version.
Point 10:Please mention the limitations of this study and suggest a few future works. Limitations could be something I mentioned earlier.
Response 10: Thanks for the comment. Considering the comment, we have mentioned the limitation of this study and suggested a future research need to be explore in the conclusion part of this study.

Round 2
Reviewer 3 Report
The authors appear to address my comments, except providing a citation to the following sentence. Please see below.
Also, please note that further and future are redundancy. Authors can replace the last two sentences with the following.
"Although not evaluated in this study, the extant studies have shown that soil NO3-N carryover could affect wheat and maize responses to applied N [80, 81]. Hence, future research should include soil NO3-N carryover in each crop growing season to meet the crop’s nutrient requirement level and maximize NUE at a given yield level in acidic soil under a wheat-maize rotation."
80. Raun, W. R., Dhillon, J., Aula, L., Eickhoff, E., Weymeyer, G., Figueirdeo, B., ... & Fornah, A. (2019). Unpredictable nature of the environment on nitrogen supply and demand. Agronomy Journal, 111(6), 2786-2791.
81. Dhakal, C., Lange, K., Parajulee, M. N., & Segarra, E. (2019). Dynamic optimization of nitrogen in plateau cotton yield functions with nitrogen carryover considerations. Journal of Agricultural and Applied Economics, 51(3), 385-401.
Author Response
Reviewer 3
Responses to comments raised by Reviewer
Manuscript ID: agronomy-1366475.
Manuscript Title: Long-term fertilization and lime-induced soil pH changes affect nitrogen use efficiency and grain yields in acidic soil under wheat-maize rotation.
Authors: Nano Alemu Daba, Li Dongchu, Huang Jing, Han Tianfu, Zhang Lu, Sehrish Ali,
Muhammad Numan Khan, Du Jiangxue, Liu Shujun, Tsegaye Gemechu Legesse, Liu
Lisheng, Xu Yongmei, Zhang Huimin and Wang Boren.
Point1: The authors appear to address my comments, except providing a citation to the following sentence. Please see below.
Also, please note that further and future are redundancy. Authors can replace the last two sentences with the following.
Response 1: Dear Reviewer, thank you very much for your constructive comment. We agree with your comment and replaced the last two sentences of our conclusion with the following.
"Although not evaluated in this study, the extant studies have shown that soil NO3--N carryover could affect wheat and maize responses to applied N [80, 81]. Hence, future research should include soil NO3--N carryover in each crop growing season to meet the crop’s nutrient requirement level and maximize NUE at a given yield level in acidic soil under a wheat-maize rotation."
Point2:
- Raun, W. R., Dhillon, J., Aula, L., Eickhoff, E., Weymeyer, G., Figueirdeo, B., ... & Fornah, A. (2019). Unpredictable nature of the environment on nitrogen supply and demand. Agronomy Journal, 111(6), 2786-2791.
- Dhakal, C., Lange, K., Parajulee, M. N., & Segarra, E. (2019). Dynamic optimization of nitrogen in plateau cotton yield functions with nitrogen carryover considerations. Journal of Agricultural and Applied Economics, 51(3), 385-401.
Response2: Thank you very much for providing us the references. Based on the Agronomy journal references list format style, we have listed them as follows.
- 80. Raun, W.R.; Dhillon, J.; Aula, L.; Eickhoff, E.; Weymeyer, G.; Figueirdeo, B.; Fornah, A. Unpredictable nature of the environment on nitrogen supply and demand. 2019, 111, 2786-2791.
- Dhakal, C.; Lange, K.; Parajulee, M.N.; Segarra, E. Dynamic optimization of nitrogen in plateau cotton yield functions with nitrogen carryover considerations. Journal of Agricultural and Applied Economics 2019,51, 385-401.

This manuscript is a resubmission of an earlier submission. The following is a list of the peer review reports and author responses from that submission.
Round 1
Reviewer 1 Report
Dear authors,
The manuscript " Long-term fertilization and lime-induced soil pH changes affect nitrogen use efficiency and grain yields in acidic soil under wheat-maize rotation" deals with a very important and from my point of view little bit forgotten topic- amelioration of acid soils. Acidification and its effect on soil properties and uptake of nutrients by plants has been well described in many scientific researches, but there is lack of long-term experiments especially under field conditions.
Part of Introduction with information’s about past results is well organized and with good number of references, but the section where hypothesis was explained is not so clear and it will be good to rewrite that part in a way to first explain problem then hypothesis and in the end possible solutions.
Measurements and laboratory analyses in section MATERIAL AND METHODS are well done described. But my biggest concern is the design of the experiment. It will be good to add experiment design in supplement material to better understand pseudo-replication because most models for statistical inference require true replication and without replication, we cannot estimation variability within a treatment. Also, variability will probably be underestimated.
Some other concerns:
- It will be good to explain how you calculated liming dosage
- Why you used 300 kg of N and why you apply only urea?
- Why you add only half of the residues?
- On what depth you incorporate liming material?
- Did you measure reserve acidity in the colloids pH in KCl or hydrolytic acidity?
Results are well organized but the Discussion part of the paper is too short and not well explained especial part with soil chemical properties. It will be good to connect liming and phosphorus availability and better explain effect of liming on SOM.
In my opinion it will be better to write two papers with that data. First before the liming experiment started and the second after limning. Then, the readers will better understand the problem and how hard is to remediate degraded soils.
Sincerely,
Reviewer
Author Response
Authors' Responses to Reviewer 1 Comments
Point 1: The manuscript “Long-term fertilization and lime-induced soil pH changes affect nitrogen use efficiency and grain yields in acidic soil under wheat-maize rotation" deals with a very important and from my point of view little bit forgotten topic- amelioration of acid soils. Acidification and its effect on soil properties and uptake of nutrients by plants has been well described in many scientific researches, but there is lack of long-term experiments especially under field conditions.
Part of Introduction with information’s about past results is well organized and with good number of references, but the section where hypothesis was explained is not so clear and it will be good to rewrite that part in a way to first explain problem then hypothesis and in the end possible solutions.
Measurements and laboratory analyses in section MATERIAL AND METHODS are well done described. However, my biggest concern is the design of the experiment. It will be good to add experiment design in supplement material to better understand pseudo-replication because most models for statistical inference require true replication and without replication, we cannot estimation variability within a treatment. Also, variability will probably be underestimated.
Response 1: First, we would like to thank you the reviewer for the detailed and constructive comments and suggestions. We have improved our manuscript according to your comments and suggestions. Based on reviewer comments and suggestions, we have revised the introduction and material and methods section as follows (changes in manuscript can be seen in revised manuscript version with track changes).
- 1. Introduction: in the introduction part, we have arranged in the sequence that statement of the problem explained first followed by the hypothesis and in the end possible solutions, (this can be seen in the revised manuscript version from Line 91 to 118).
- Material and methods: We thank the reviewer for this important criticism. We understand reviewer concern regarding to the experimental design that pseudo-replication can affect variability within a treatment. The experimental design of this study was established with two true replications in 1990 (which can be considered as one of its weakness). Both true replications for each treatment were treated separately. To have three sample size (n=3), we used the samples collected from plots in one of the original replicates (replication 2 in our case) as a third replication (which was a pseudo-replication). This was based the previous references, Qaswar et al., 2020 and Hurlbert., 1984. Here, Qaswar et al., 2020 were did their field experiment in the same site with our current field experiment using the same experimental design, where many of our current authors were also co-authors of this paper . As we have already mentioned, type I error can be increased due to this pseudo-replication. Similarly, (ZHANG et al. 2011; YANG, SUN, and ZHANG 2014) underlined that the weakness of this experiment was its design with only two replications. Based on reviewer comments, we have added further detailed information regarding to this issue. Changes can be seen in the revised manuscript (Lines 148 to 161)
Point 2: Some other concerns:
Response 2: Dear reviewer, thank you very much for your kind comments and suggestions. As per your comments, we have corrected all your concerns as follows:
Point 2.1: It will be good to explain how you calculated liming dosage
Response 2.1: The liming dosage was calculated based on soil pH at the beginning of the experiment, soil pH in 2010 and soil buffer capacity. We have explained these in the revised manuscript (Lines 142 and 143)
Point 2.2: Why you used 300 kg of N and why you apply only urea?
Response 2.2: The fertilizer level was calculated based on the initial test for soil nutrient levels and crop nutrient requirements in 1990. So, 300 kg of N also based on the initial soil test for crop nutrient requirement (revised manuscript Line 165). The reason why only N applied is due to urea was widely used by local farmers as a source of N while the experiment was established.
Point 2.3: Why you add only half of the residues?
Response 2.3: It was determined by the current agricultural situation. The reason is that before 2000, China's fertilizer industry had not yet met the national demand for agricultural fertilizer. However, agricultural research results show that when straw is returned to the field, microorganisms absorb part of soil nitrogen in the early stage of straw decay, and compete for nitrogen with crops, which affects the early stage of crop growth. Therfore, the main the main reason is that the ratio of carbon to nitrogen of straw is too high. Therefore, straw returning to the field should not be too large, half of the straw should be returned to the field (revised manuscript Line 169).
Point 2.4: On what depth you incorporate liming material?
Response 2.4: 0-15 cm (revised manuscript Line 143)
Point 3: Results are well organized but the Discussion part of the paper is too short and not well explained especial part with soil chemical properties. It will be good to connect liming and phosphorus availability and better explain effect of liming on SOM.
Response 3: Dear reviewer thanks for your suggestion and comments. We added more explanation to the discussion part under soil chemical properties linking with liming and phosphorus availability and effect of liming on SOM was also explained (revised manuscript version Lines 470 to 477 and Lines 517 to 524).
Point 3: In my opinion it will be better to write two papers with that data. First before the liming experiment started and the second after limning. Then, the readers will better understand the problem and how hard is to remediate degraded soils.
Response 3: Dear reviewer, thanks for your kind suggestion. Yes of course. But our intention was to compare the effect of liming within liming period in reference to before liming experiment on the same paper so that the reader simultaneously understand and even compare the effect of treatments on given parameters in long-term either with or without liming. So, hence our data organized in the way we have mentioned above it is better to use for one paper, just in the current form.

Reviewer 2 Report
The main strength of the present manuscript is the duration of the experimental study. Agricultural experiments spanning almost 30 years are rare, and they certainly lend credibility to results obtained from the manipulation of soil and crop properties, particularly in those instances where agricultural systems might respond slowly to environmental change. It should thus be of interest to readers of Agronomy.
However, some of the methods need some clarification, especially in relation to the split plot design. At its current version, the authors provide no information on sample size (number of plots), which makes it difficult to contextualise the results of the statistical analysis since we do not know the degrees of freedom of the variance models used. Tables and figures in the Results section also need improvements. The Discussion needs some editing; the authors seem to switch between their results and those of other studies but that is not coming across clearly. I would suggest rewriting several passages to make that clearer.
I make several suggestions below to improve the quality of the manuscript (by line number):
Abstract
L18: by ‘improves’ you mean ‘increases’? I’d suggest being more specific.
L20: instead of ‘crops GY’, you could specify what crops you are looking at.
L22-24: I’d suggest ‘… carried out between 1991-2010 (before liming) and 2011-2018 (after liming).’ Also, I’m not convinced BLE is a good abbreviation to use. You could simply state ‘before liming’ throughout the text.
L26-28: this sentence lacks some context. It is not clear at this point what negative effects are. Instead, you could simply present the main results and draw some conclusions towards the end of the Abstract. Also, you should specify what ‘selected soil chemical properties’ are.
L29-31: this sentence is poorly constructed, please rephrase it.
L38-40: the meaning of this sentence is not clear, please rewrite it.
Introduction
L45: is it ‘assumed’, or is there enough evidence to support your statement (‘recognised’?)? I suggest rewording it.
L50: at no point in the text you specify what is meant by acidification. To what pH threshold exactly are you attributing acidic conditions? It would be helpful to contextualise what acidic means in the context of wheat-maize cropping. Different systems might have different definitions of acidic.
L56: I find the beginning of this sentence odd. pH is an intrinsic quality of soils and not induced by fertilisation. Do you mean to say ‘changes in soil pH induced by fertilisation’?
L59: … severely acidic (i.e., below x pH)? pH improvement = pH increase?
L68-69: duration of liming?
L70: not clear what is meant by ‘longevity’ here. Please clarify.
L81: define the NUE of wheat-maize systems?
L86: the bracketed percentage interval should come after REN.
L89-91: any evidence to back up this claim?
Methods
L123: what soil chemical properties are these? You must specify.
L126: you should specify how many plots received the liming treatment.
L132-137: this paragraph is very confusing. I do not understand what is meant by ‘A pseudo-replication was used as a third replication…’. Pseudo-replication is an inherent property of split-plot designs, which is normally dealt with through appropriate statistical testing. How exactly did you address the issue of pseudo-replication? Also, you should use this paragraph to clarify how many plots of each treatment you had and the first order and second order treatments.
L159: from each treatment?
L200-202: and where are the results of the statistical power analysis. Did you conclude the study was large enough?
L203-207: I find it odd to characterise variables as explanatory and response if you are testing correlations in ordination space. I would suggest revisiting this terminology; I am not convinced it is appropriate.
Results
L210-234: the text and the table seem to use different starting dates for the first period of the experiment, with some starting in 1990 and others starting in 1991. Be consistent throughout the text and tables.
Table 1: what are the small letters after each value? Differences between means after ANOVA? If so, the table caption should specify, and also include what statistical analysis was used. Also, if these are mean values, can you include a measure of variability around them (e.g., standard deviation)?
Table 2: similar comments to Table 1 above. In addition, what are the ‘Mean effect’ values? Statistical tests? If so, the table caption should specify.
L240-253: are the results presented in this paragraph solely related to Fig. 1? If so, I’d suggest you introduce Fig. 1 first and in the second paragraph Fig. 2. Mentioning both at the beginning of the paragraph makes it difficult to place where the results are illustrated.
L244-247: if these are mean values, please provide a measure of variability.
Fig. 2: the caption should make clear that these differences are in relation to control plots.
L272-283: these seem to be important results, but this analysis was not introduced in the ‘Statistical analysis’ section in the Methods.
L275: Instead of saying ‘longer-term’, you could say ‘during the experimental period’. Longer-term can be interpreted in different ways.
Fig. 3: the panels in the figure have no letters (a to c). Also, the figure caption seems to present the results in different order to the one presented in the figure (Al, Mg, Ca).
Fig. 4: personally, I am not a big fan of 3D plots. It is difficult to get a proper sense of the results. Is there an alternative approach to this?
L299-318: please include measures of variability around any mean values presented here.
L301: again, I think BLE is an odd abbreviation. You could simply say ‘before liming’. I’d suggest you revisit this throughout the text.
Fig. 5: please state what statistical analysis the p value refers to.
L326-330: what does that mean then?
L358-372: I’d suggest you separate the results in Table 3 from those in Table 4. Specify at the end of each sentence in which table to find those results.
Table 3 and Table 4: please specify what statistical analysis was used for comparisons in the table captions. Table 4 caption states that N uptake is shown in %, but the table indicates kg ha-1. Also, if these values are means, can you include a measure of variability?
Fig. 7: the years along the x axes seem wrong. The last year in the middle panels is 2010; should it be 2019?
L397: superiority in terms of what exactly? Improving crop NUE?
Discussion
L407: not sure what is meant by ‘exhausted’ here. What is ‘exhausted’ soil pH?
L410: can you clarify after how long liming started having an effect on these soil properties? Long-term seems a bit vague.
L413: again, how long is long-term here?
L446-448: my impression from your results is that liming interacts with N addition to ameliorate soil acidification. This sentence comes across as if reduced soil acidification was due to N application alone, when liming is probably the main driver here.
L450: lower or higher C/N ratio? This seems to contradict the beginning of the sentence. This sentence is confusing, please rephrase.
L463-465: this sentence is also confusing. Low soil pH leads to acidification, not the other way round.
L471-473: please clarify you are referring to the results of other studies. Are they directly comparable to yours? Same crop? Similar site conditions?
L479-481: I’d imagine that was one of the reasons, but not the sole reason.
L519-520: what sort of difference between long-term and short-term are you referring to here? Five years? Ten years? Please be more specific.
L521-522: not sure what is meant by ‘liming years’ here. Does this refer to the frequency of lime application over a period of time? Or the amount of lime applied within certain years? Please clarify.
L529-531: again, are these the results from another study or yours? It is not clear enough.
L543: is it fair to say liming interacts with N addition to improve crop yield and NUE? I’d suggest rephrasing this sentence.
L547: What is long-term? In the following sentence you seem to suggest eight years is not long enough, so how long would be long enough for changes to have a beneficial effect?
General: I am not convinced you need to constantly refer to tables and figures in the Discussion section.
Conclusions
L553: specify for how long liming took place in your study.
L559-560: is there a minimum number of years before benefits can be measured? If this is a recommendation, perhaps you should try to be more specific.
Author Response
Authors' Responses to Reviewer 2 Comments
Point 1: The main strength of the present manuscript is the duration of the experimental study. Agricultural experiments spanning almost 30 years are rare, and they certainly lend credibility to results obtained from the manipulation of soil and crop properties, particularly in those instances where agricultural systems might respond slowly to environmental change. It should thus be of interest to readers of Agronomy.
However, some of the methods need some clarification, especially in relation to the split plot design. At its current version, the authors provide no information on sample size (number of plots), which makes it difficult to contextualise the results of the statistical analysis since we do not know the degrees of freedom of the variance models used. Tables and figures in the Results section also need improvements. The Discussion needs some editing; the authors seem to switch between their results and those of other studies but that is not coming across clearly. I would suggest rewriting several passages to make that clearer.
I make several suggestions below to improve the quality of the manuscript (by line number):
Response 1: We thank the reviewer for these important comments, suggestions and criticism, which are very helpful to further improve our manuscript. We have now revised the manuscript according your comments and suggestions. Hence, all the comments raised in the above general comments are also mentioned below under specific comments; we would like provide our point-by-point responses below under each specific comments or suggestions (by line number)
Abstract
Point 1: L18: by ‘improves’ you mean ‘increases’? I’d suggest being more specific.
Response 1: We thank the reviewer for pointing this out. It should be increases. We have corrected it (by changing the word ‘improves’ to ‘increases’ (revised manuscript line 18).
Point 2: L20: instead of ‘crops GY’, you could specify what crops you are looking at.
Response 2: We thank the reviewer for this important suggestion. We have changed ‘crops GY’ to wheat and maize GY (revised manuscript line 21)
Point 3: L22-24: I’d suggest ‘… carried out between 1991-2010 (before liming) and 2011-2018 (after liming).’ Also, I’m not convinced BLE is a good abbreviation to use. You could simply state ‘before liming’ throughout the text.
Response 3: We thank the reviewer for this important suggestion. We have made corrections according to your comment. The correction can be checked in track changes, revised manuscript (Lines 23-24). We have stated before liming instead of “BLE” abbreviation or “before the liming experiment” throughout the text.
Point 4: L26-28: this sentence lacks some context. It is not clear at this point what negative effects are. Instead, you could simply present the main results and draw some conclusions towards the end of the Abstract. Also, you should specify what ‘selected soil chemical properties’ are.
Response 4: We thank the reviewer for this important criticism. As per your comment, the main results were presented instead of what stated in the previous version (revised manuscript Lines 26-29). The ‘selected soil chemical pro(soil pH, NH4+-N, TN, AP, exchangeable soil Al3+ and Ca2+),. The conclusion also drawn to the end of the conclusion perties’ are also specified as (revised manuscript lines 26 and 27)
Point 5: L29-31: this sentence is poorly constructed, please rephrase it.
Response 5: We thank the reviewer for pointing this out. We have rephrased the sentence. The changes can be found in track change manuscript version (revised manuscript lines 30-33).
Point 6: L38-40: the meaning of this sentence is not clear, please rewrite it.
Response 6: We have corrected this sentence as per your comment and the changes can be found in track changes (revised manuscript lines 40- 46)
Introduction
Point 7: L45: is it ‘assumed’, or is there enough evidence to support your statement (‘recognised’?)? I suggest rewording it.
Response 7: We thank the reviewer for pointing this out. We have changed the “assumed” word to ‘recognised’ based on your comment (revised manuscript line 51)
Point 8: L50: At no point in the text, you specify what is meant by acidification. To what pH threshold exactly are you attributing acidic conditions? It would be helpful to contextualise what acidic means in the context of wheat-maize cropping. Different systems might have different definitions of acidic.
Response 8: Dear reviewer, thanks for your important comment, criticism and suggestion. We have added specific definition of acidification in line with wheat-maize system and revised it as per your suggestion (revised manuscript lines 58 to 63).
Point 9: L56: I find the beginning of this sentence odd. pH is an intrinsic quality of soils and not induced by fertilisation. Do you mean to say ‘changes in soil pH induced by fertilisation’?
Response 9: We thank the reviewer for pointing this out. Yes, this sentence should be corrected as the reviewer suggested. Therefore, we have changed it to ‘changes in soil pH induced by fertilisation’ (revised manuscript Line 64).
Point 10: L59: … severely acidic (i.e., below x pH)? pH improvement = pH increase?
Response 10: Thank you for your kind suggestion. We have improved this part by including the pH value in referring acidic level and pH improvement also changed to pH increase (revised manuscript Line 67). Dear reviewer, here we would like to inform you that we have replaced pH increases with pH improvement throughout the text, as the term pH improvement is inappropriate to refer pH increase.
Point 11: L68-69: duration of liming?
Response 11: Thank you for the correction. Yes, it is duration of liming. So, we have modified this phrase as duration of liming (revised manuscript Line 79).
Point 12: L70: not clear what is meant by ‘longevity’ here. Please clarify.
Response 12: This is to refer the pH change increases continuously over years in limed soil. But due to the word seems somewhat confusing and the sentence is self-explanatory to indicate our intention, we have deleted this word from the sentence (revised manuscript line 81)
Point 13: L81: define the NUE of wheat-maize systems?
Response 13: Thank you for your kind suggestion. We have corrected this sentence according to your suggestion (revised manuscript Line 91)
Point 14: L86: the bracketed percentage interval should come after REN.
Response 14: Thanks for the comment. We have corrected this sequence as per your comment (revised manuscript Line 96)
Point 15: L89-91: any evidence to back up this claim?
Response 15: We have included the reference to this part (revised manuscript Line 102)
Methods
Point 16: L123: what soil chemical properties are these? You must specify.
Response 16: Dear reviewer thanks for your suggestion. We have specified the soil chemical properties as soil pH, soil available K, soil available P, NO3- -N, NH4+-N, SOM, total N, C: N ratio and soil changeable cations (Al3+, Ca2+ and Mg2+) ( revised manuscript lines 133-135).
Point 17: L126: you should specify how many plots received the liming treatment.
Response 17: We thank the reviewer for this important comment. In this experiment, there were 12 plots receiving lime over 2011 to 2018. We have included the detailed information in material and methods of revised manuscript (lines 148 to 150)
Point 18: L132-137: this paragraph is very confusing. I do not understand what is meant by ‘A pseudo-replication was used as a third replication…’. Pseudo-replication is an inherent property of split-plot designs, which is normally dealt with through appropriate statistical testing. How exactly did you address the issue of pseudo-replication? Also, you should use this paragraph to clarify how many plots of each treatment you had and the first order and second order treatments.
Response 18: We thank the reviewer for this important criticism. We understand your concerns here. To make it clear, the experimental design of this study was established with two true replications in 1990 (which can be considered as one of its weakness). So, to have the standard replications for testing treatments effect or three sample size (n=3) for each treatment we have used pseudo-replication using the samples collected from one of our true replications (i.e. replication 2 in our case), as previously described by Qaswar et al. (2020) and Hurlbert, (1984). Here, Qaswar et al., 2020, did their experiment at the same experimental site with similar experimental design under two replications using psued-replication as third replication. Similarly, (ZHANG et al. 2011; YANG, SUN, and ZHANG 2014) in their previous reports from this experimental site underlined that, the weakness of this experiment was its design with only two true replications. We have added further detailed information regarding to this issue in the revised manuscript (lines Lines 148 to 161).
Point 19: L159: from each treatment?
Response 19: Thanks for the comment. Yes, it should be from each treatment. We have corrected (revised manuscript line 185)
Point 20: L200-202: and where are the results of the statistical power analysis. Did you conclude the study was large enough?
Response 20: We thank the reviewer for pointing this out. However, after rechecked this statement, we found that it is inappropriate and therefore we have excluded it from the section.
Point 21: L203-207: I find it odd to characterise variables as explanatory and response if you are testing correlations in ordination space. I would suggest revisiting this terminology; I am not convinced it is appropriate.
Response 21: Thank you so much for the important suggestion. We have modified the terminology based on your recommendation (revised manuscript Lines 228-231).
Results
Point 22: L210-234: the text and the table seem to use different starting dates for the first period of the experiment, with some starting in 1990 and others starting in 1991. Be consistent throughout the text and tables.
Response 22: Thank you for the correction. 1990 was the year that this experiment established. The fertilization experiment started in 1991. Therefore, it should be 1991- 2010. We have corrected it (revised manuscript line 257)
Point 23: Table 1: what are the small letters after each value? Differences between means after ANOVA? If so, the table caption should specify, and also include what statistical analysis was used. Also, if these are mean values, can you include a measure of variability around them (e.g., standard deviation)?
Response 23: We thank the reviewer for this important suggestion. Means followed by different letters are significantly (p ≤ 0.05) different from each other according to Tukey’s HSD test (revised manuscript caption of Table 1). Dear reviewer, regarding to include a major of variability, the editor suggested us to delete the SD values from the body of tables. The editor justified that “these values are not required because the lettering system is used to separate treatment effects”. However, we can include the SD again if the editor agrees.
Point 24: Table 2: similar comments to Table 1 above. In addition, what are the ‘Mean effect’ values? Statistical tests? If so, the table caption should specify
Response 24: Dear respected editor, we have similar responses to table 1 above. The mean effect in this table refers to the mean values of all treatments with liming (+L) and without liming (-L). * and ** refers the mean effect is significant at p p ≤0.05 and p≤0.01, respectively according to Tukey’s HSD test (revised manuscript caption of Table 2).
Point 25: L240-253: are the results presented in this paragraph solely related to Fig. 1? If so, I’d suggest you introduce Fig. 1 first and in the second paragraph Fig. 2. Mentioning both at the beginning of the paragraph makes it difficult to place where the results are illustrated.
Response 25: We thank the reviewer for this important suggestion. Yes, we have introduced only Fig. 1 in this paragraph and Fig. 2 in the second paragraph according to your suggestion (revised manuscript lines 276-289)
Point 26: L244-247: if these are mean values, please provide a measure of variability.
Response 26: We thank the reviewer for the suggestion. We can include the measure of variability, but our concern here is that it will increase the total word count/pages of paper. Because, if once we start to include SD or SE for values of one parameter in the text body, it may needed also for others to be consistent throughout the text.
Point 27: Fig. 2: the caption should make clear that these differences are in relation to control plots.
Response 27: We thank the reviewer for this important comment. We have specified it in the figure caption (revised manuscript Figure.2).
Point 28: L272-283: these seem to be important results, but this analysis was not introduced in the ‘Statistical analysis’ section in the Methods.
Response 28: We thank the reviewer for this important comment. We have introduced them in Statistical analysis’ (revised manuscript lines 226-228).
Point 29: L275: Instead of saying ‘longer-term’, you could say ‘during the experimental period’. Longer-term can be interpreted in different ways.
Response 29: We thank the reviewer for this important suggestion. We have replaced longer-term’ with ‘during the experimental period (revised manuscript line 306)
Point 30: Fig. 3: the panels in the figure have no letters (a to c). Also, the figure caption seems to present the results in different order to the one presented in the figure (Al, Mg, Ca).
Response 30: We thank the reviewer for this important correction. We have assigned the letter (a to c) to the panels in the figure (Fig. 3). We have also corrected the order of (Al, Mg, Ca) in this figure (revised manuscript figure 3).
Point 31: Fig. 4: personally, I am not a big fan of 3D plots. It is difficult to get a proper sense of the results. Is there an alternative approach to this?
Response 31: We thank the reviewer for pointing this out. We have presented these results in other form of figure that gives good sense in presenting the results (revised manuscript Figure 4)
Point 32: L299-318: please include measures of variability around any mean values presented here.
Response 32: We thank the reviewer for the comment. For this suggestion, we would like to use the response we have provided under “response 26” above.
Point 33: L301: again, I think BLE is an odd abbreviation. You could simply say ‘before liming’. I’d suggest you revisit this throughout the text.
Response 33: We thank the reviewer this interesting for suggestion. Thank you for your good suggestion. We replaced “BLE” with “before liming” throughout the text.
Point 34: Fig. 5: please state what statistical analysis the p value refers to.
Response 34: We thank the reviewer for pointing this out. We have corrected this one in table caption (revised manuscript Figure 5)
Point 35: L326-330: what does that mean then?
Response 35: We thank the reviewer for pointing this out. This explaining variation and separation of fertilizer treatments in -L and +L along PC1 and PC2. The angle between the arrows in the biplot represents the correlation between variables with 0 and 180° indicating the maximum positive and negative correlation and 90° indicating no correlation. We have clarified these in the revised manuscript (lines 366-369)
Point 36: L358-372: I’d suggest you separate the results in Table 3 from those in Table 4. Specify at the end of each sentence in which table to find those results.
Response 36: We thank the reviewer for this important suggestion. We have separated them as per your comment (revised manuscript lines 383-397)
Point 37: Table 3 and Table 4: please specify what statistical analysis was used for comparisons in the table captions. Table 4 caption states that N uptake is shown in %, but the table indicates kg ha-1. Also, if these values are means, can you include a measure of variability?
Response 37: We thank the reviewer for this comment. We have specified the statistical analysis and corrected the unit of N uptake as kg ha-1 in the table caption. Measure of variability also included (Tables 3 and 4)
Point 38: Fig. 7: the years along the x axes seem wrong. The last year in the middle panels is 2010; should it be 2019?
Response 38: We thank the reviewer for this important correction. We have clarified more the years along the x axes. The year in the middle panel (2010) is correct, but hence it may make confusion for the reader, we have revised this arrangement (revised manuscript figure 7)
Point 39: L397: superiority in terms of what exactly? Improving crop NUE?
Response 39: We thank the reviewer for this kind suggestion. Yes, exactly it was in improving crop NUE. So have corrected it in line with your suggestion (revised manuscript line 425)
Discussion
Point 40: L407: not sure what is meant by ‘exhausted’ here. What is ‘exhausted’ soil pH?
Response 40: We thank the reviewer for this important comment. Here, by “exhausted” we want to refer the soil depletion. However, it not seems an appropriate term to refer soil depletion, thus we have changed this term and revised the entire of this paragraph (revised manuscript lines 470-477)
Point 41: L410: can you clarify after how long liming started having an effect on these soil properties? Long-term seems a bit vague.
Response 41: We thank the reviewer for this comment. We have specified the “long-term” to “7-years” (revised manuscript line 479)
Point 42: L413: again, how long is long-term here?
Response 42: We have specified this this one also to 7 years liming (revised manuscript line)
Point 43: L446-448: my impression from your results is that liming interacts with N addition to ameliorate soil acidification. This sentence comes across as if reduced soil acidification was due to N application alone, when liming is probably the main driver here.
Response 43: Dear reviewer, thank you so much for your kind comment. The interpretation of the result in this sentence was wrong (not in line with our results), and we have corrected according to our results (revised manuscript lines 515 and 516).
Point 44: L450: lower or higher C/N ratio? This seems to contradict the beginning of the sentence. This sentence is confusing, please rephrase.
Response 44: Thank you for your correction. We have checked this sentence, and it should be higher. Therefore, we have corrected accordingly (revised manuscript line 526).
Point 45: L463-465: this sentence is also confusing. Low soil pH leads to acidification, not the other way round.
Response 45: Thank you for the correction. Yes, it is true the lower the pH of soil, the greater the acidity or low soil pH leads to acidification. We have corrected in the text also in line with this concept (revised manuscript lines 539 and 540).
Point 46: L471-473: please clarify you are referring to the results of other studies. Are they directly comparable to yours? Same crop? Similar site conditions?
Response 46: Thank you for the comment. Yes, directly comparable with our study (the same crop and site). We clarified more in the text (revised manuscript line 547)
Point 47: L479-481: I’d imagine that was one of the reasons, but not the sole reason.
Response 47: Thank you for the comment. Yes, it is not the only reason, so we have modified this sentence (revised manuscript line 555)
Point 48: L519-520: what sort of difference between long-term and short-term are you referring to here? Five years? Ten years? Please be more specific.
Response 48: Thank you for the comment. Here, short-term is referring to < 5 years, while long-term is referring to ≥ 5 years. We have revised in the text too in line with this idea (revised manuscript line 596)
Point 49: L521-522: Not sure, what is meant by ‘liming years’ here. Does this refer to the frequency of lime application over a period? or the amount of lime applied within certain years? Please clarify.
Response 49: This is to mean the duration of liming or referring to the time (in year) after liming. So, due to liming year is a little bit confusing, we have changed it to duration of liming (revised manuscript line 597)
Point 50: L529-531: again, are these the results from another study or yours? It is not clear enough.
Response 50: Thank you for your correction. These results are ours, and we have corrected it now (revised manuscript line 607).
Point 51: L543: is it fair to say liming interacts with N addition to improve crop yield and NUE? I’d suggest rephrasing this sentence.
Response 51: Thank you for the comment. These sentences were not articulated well. Now we have rephrased them (revised manuscript revised manuscript lines 618-619).
Point 52: L547: What is long-term? In the following sentence you seem to suggest eight years is not long enough, so how long would be long enough for changes to have a beneficial effect?
Response 52: Thank you so much for your comment. Here, our base was 5 years to categorize a given experiment either to short- or long-term. In our case, the liming period was undergoing for 8 years. Comparing the results of corresponding treatments under –L and +L over eight years, most tested parameters showed significant change. However, these changes (particularly GY and NUE) not at their potential or maximum values after 8 years, indicating that further years required for liming. At this moment, it is difficult to estimate the exact enough periods (long enough) to have potential or maximum values for those parameters. We hope, may be this issue will addressed in the future by scholars conducting similar experiment.
Point 53: General: I am not convinced you need to constantly refer to tables and figures in the Discussion section.
Response 53: Thank you for the comment. Just based your comment, we have removed some figures and tables that repeatedly cited in the discussion part (the removed cited figures and tables can be seen in the track changes version)
Conclusions
Point 54: L553: specify for how long liming took place in your study.
Response 54: Thank you for the comment. We have specified this to 8-years (revised manuscript line 628)
Point 55: L559-560: is there a minimum number of years before benefits can be measured? If this is a recommendation, perhaps you should try to be more specific.
Response 55: Thank you for the comment. We have specified here too (revised manuscript line 634)
